# The trajectory of cortical GABA across the lifespan, an individual participant data meta-analysis of edited MRS studies

Eric C Porges[1,2,3]*, Greg Jensen[4,5], Brent Foster[1,2,3], Richard AE Edden[6,7], Nicolaas AJ Puts[6,7,8]

[1]Center for Cognitive Aging and Memory, University of Florida, Gainesville, United States; [2]McKnight Brain Research Foundation, University of Florida, United States, United States; [3]Department of Clinical and Health Psychology, University of Florida, Gainesville, United States; [4]Department of Neuroscience, Columbia University Medical Center, New York, United States; [5]Zuckerman Mind Brain Behavior Institute, Columbia University, New York, United States; [6]Russell H. Morgan Department of Radiology and Radiological Science, The Johns Hopkins University School of Medicine, Baltimore, United States; [7]F.M. Kirby Research Center for Functional Brain Imaging, Kennedy Krieger Institute, Baltimore, United States; [8]Department of Forensic and Neurodevelopmental Sciences, Sackler Institute for Translational Neurodevelopment, Institute of Psychiatry, Psychology, and Neuroscience, King's College London, London, United Kingdom

**Abstract** γ-Aminobutyric acid (GABA) is the principal inhibitory neurotransmitter in the human brain and can be measured with magnetic resonance spectroscopy (MRS). Conflicting accounts report decreases and increases in cortical GABA levels across the lifespan. This incompatibility may be an artifact of the size and age range of the samples utilized in these studies. No single study to date has included the entire lifespan. In this study, eight suitable datasets were integrated to generate a model of the trajectory of frontal GABA estimates (as reported through edited MRS; both expressed as ratios and in institutional units) across the lifespan. Data were fit using both a log-normal curve and a nonparametric spline as regression models using a multi-level Bayesian model utilizing the Stan language. Integrated data show that an asymmetric lifespan trajectory of frontal GABA measures involves an early period of increase, followed by a period of stability during early adulthood, with a gradual decrease during adulthood and aging that is described well by both spline and log-normal models. The information gained will provide a general framework to inform expectations of future studies based on the age of the population being studied.

*For correspondence:
eporges@phhp.ufl.edu

**Competing interests:** The authors declare that no competing interests exist.

## Introduction

Magnetic resonance spectroscopy (MRS) is a non-invasive imaging technique that allows for the measurement of levels of metabolites. Of particular interest to the neurosciences is the measurement of specific neurotransmitters such γ-aminobutyric acid (GABA) in vivo (*Edden and Barker, 2007*; *Mescher et al., 1998*; *Mullins et al., 2014*; *Puts et al., 2011*; *Rothman et al., 1993*). GABA is the main inhibitory neurotransmitter in the human nervous system and plays a fundamental role in central nervous system function (*Buzsáki et al., 2007*). A number of studies have explored the relationship between cortical GABA (as measured with MRS) and age in various contexts. These studies have found that aging-related changes in GABA are consistently associated with cognitive and

neurophysiological outcomes that change across the lifespan. These findings have important implications for both healthy and pathological development and aging.

Most prior studies of aging-related differences in GABA have utilized a restricted age range. Some have reported increases in GABA as age increases in left frontal cortex between 13 and 53 years (*Ghisleni et al., 2015*). Others have reported decreases in medial frontal GABA (*Marenco et al., 2018*, 18–55 years; *Porges et al., 2017a*, 43–92 years; *Rowland et al., 2016*, 16–62 years) and GABA/creatine plus phosphocreatine (Cr+PCr) (*Gao et al., 2013*, 20–76 years). Gao and Porges show the same aging-related decrease in midline parietal GABA/Cr+PCr and GABA, respectively, as does (*Simmonite et al., 2019*) for GABA/Cr+PCr in occipital regions (18–87 years). Still others reported no significant aging-related changes in GABA (*Aufhaus et al., 2013*, 21–53 years) or GABA/Cr+PCr (*Mikkelsen et al., 2017*, 18–48 years). This inconsistency makes the results difficult to interpret, as partially overlapping age ranges produce conflicting trajectories. For example, the age range of participants reported by *Gao et al., 2013* has substantial overlap with those reported by *Ghisleni et al., 2015*, yet has apparently conflicting trajectories. Of interest, all studies were cross-sectional rather than looking at within-subject *change* in GABA with age.

However, conflicting results only exist when looking at different age ranges and consistent findings have been shown in studies of similar age range. For example, both *Porges et al., 2017a* and *Gao et al., 2013* focus on adults through advanced age and both report aging-related decrease in GABA. Here, we predict that this apparent conflict in the relationship between GABA and age in the literature is the result of restricted age ranges within each study and that the various linear relationships are part of larger, non-linear asymmetric lifespan trajectory of GABA with age. This would be consistent with prior work showing non-linear relationships with age across the human lifespan including inhibition-dependent behavior (*Williams et al., 1999*), GAD65 expression (*Pinto et al., 2010*), cortical thinning (*Fjell et al., 2019*; *Walhovd et al., 2016*; *Raznahan et al., 2011*), and white matter development (*Lebel et al., 2012*). With respect to MRS, this prediction of a non-linear asymmetric lifespan trajectory, with an increase during childhood and aging-associated gradual decrease, is further suggested by an exploratory LOESS (locally estimated scatterplot smoothing) regression for macromolecule-suppressed medial-frontal GABA × age in healthy controls reported by *Rowland et al., 2016*. Based on this prior work, and that of individual study of the relationship between GABA and age, when collating data across studies, we predict to find a non-linear aging-related trajectory involving an increase of GABA in childhood, relative stability in adolescence, followed by a gradual decrease during adulthood. To date, no single study has explored the lifespan trajectory of cortical GABA spanning development, adulthood, and aging. In the absence of a lifespan study, we implemented an individual participant data meta-analytic (IPD-MA) approach following Preferred Reporting Items for Systematic Reviews and Meta-Analyses (PRISMA) guidelines (*Moher et al., 2009*; see Figure 5) supplemented with data collected by the authors and previously published in summary form (*Puts et al., 2017*).

The majority of MRS studies of GABA at 3 T have utilized J-difference editing to selectively 'edit' the GABA signal (e.g. *Mullins et al., 2014*). Editing is necessary at 3 T due to the low concentration of GABA in the human brain (1–2 mM) (*Harris et al., 2017*). In unedited MRS, signal from higher concentration molecules like *N*-acetylaspartate (NAA) and Cr+PCr masks the GABA signal. The most widely used J-difference editing MRS technique is MEGA-PRESS (*Mescher et al., 1998*), in which a GABA-selective editing pulse at 1.9 ppm is applied in half of the experiment (edit-ON and is coupled to a GABA signal at 3 ppm) but not in the other half (edit-OFF, where ON and OFF acquisitions are typically interleaved). The difference spectrum (ON-OFF acquisitions) shows only those signals affected by the frequency-selective editing pulse, revealing a GABA signal at 3 ppm. The difference spectrum is further complicated by a macromolecule signal at 3 ppm which is coupled to another macromolecule signal at 1.7 ppm and thus falls within the envelope of the editing pulse, resulting in a co-edited macromolecule signal as part of the 3 ppm GABA signal (*Edden et al., 2014*; *Henry et al., 2001*). Consequently, most studies refer to the GABA signal as 'GABA+'. Both macromolecule-suppressed and GABA+ measures are included here (Table 2). Due to its low concentration, measurement of the GABA-edited signal in humans requires a large voxel (most commonly 27 cm$^3$; *Mullins et al., 2014*; *Peek et al., 2020*) to keep acquisition times reasonable (~10 min) and to provide an adequate signal-to-noise ratio (SNR) (*Mikkelsen et al., 2018*). This limitation constrains the spatial specificity of the measurement to coarse regions that often lack discrete functional specificity.

The functional relevance of MRS measures of GABA is important to note. Measured GABA levels include both intracellular (both somatic and synaptic) and extracellular contributions to the overall GABA concentration. However, these relative contributions are not well known; GABA levels measured via MRS at rest describe a physiological characteristic of the tissue measured and, while associated with functional metrics (e.g. neurophysiological response or behavior), are better interpreted in a manner similar to structural neuroimaging. However, studies have suggested that the majority of the GABA measured using MRS reflects intracellular somatic levels rather than synaptic or extracellular levels since MRS measures of GABA were associated with expression of the 67 isoform of glutamic acid decarboxylase (GAD) which is predominantly present in the soma of the neuron (*Marenco et al., 2011*; *Rae, 2014*; *Stagg et al., 2011*). Other than that, no study has shown clear discrimination between contributions of different pools to the GABA signal. As such, the GABA signal is considered to be reflective of inhibitory tone (*Rae, 2014*). GABA levels measured via MRS in young adults at rest have been reported to be stable for up to 7 months (*Near et al., 2014*) and do not exhibit a diurnal rhythm (*Evans et al., 2010*).

In this manuscript, we statistically combine datasets of published research that used MEGA-PRESS to measure GABA in discrete age ranges where individual data points were presented relative to age to present a non-linear model for GABA estimates over the human lifespan using an individual participant data approach. This work is motivated by several studies showing that MEGA-PRESS measures of cortical GABA are relevant to both development and aging with a specific emphasis on cognition and perception. We further discuss the lack of available data across the lifespan.

## The importance of GABA in the context of aging in health and disease

GABA as measured with MRS has been linked to clinical and cognitive outcomes. Alterations of GABA levels are seen in neurodevelopmental disorders such as ADHD (*Bollmann et al., 2015*; *Edden et al., 2012*), autism spectrum disorder (*Cochran et al., 2015*; *Drenthen et al., 2016*; *Gaetz et al., 2014*; *Puts et al., 2017*), and Tourette syndrome (*Puts et al., 2015*), as well as in other neurological and psychiatric disorders. These clinical differences are reviewed by *Puts and Edden, 2012* and by *Schür et al., 2016*. Associations with GABA measures have been reported across a variety of sensory and cognitive domains (for a review, see *Duncan et al., 2014*) associations with other measures of brain function (for a review, see *Duncan et al., 2014*). Given the functional relevance of GABA in the context of both pathological and healthy cohorts (especially in the context of development and aging), understanding how GABA changes with age in a healthy cohort is important to understand typical and atypical progression of behavior. However, few studies have included the impact of age to their investigations. Our approach allows for a systematic review of existing work studying GABA across development and aging with immediate impact on future studies in the context of development and aging.

## GABA across the lifespan

To date, cross-sectional and longitudinal investigations of cortical GABA across the *entire* human lifespan have yet to be published. However, there have been recent reports investigating the relationship between cortical GABA levels and discrete age ranges in humans that test a linear association focused on a specific population (e.g. 'aging'). As discussed above, using these studies to make blanket statements about correlations between age and GABA is inappropriate as these studies report a variety of effects. Some suggest a positive correlation between GABA and age, others hint at a negative correlation, and still others conclude that there is no relationship at all. Here, we discuss these reports in the context of human development and divide them into three categories: developmental, adult, and aging.

### Developmental

The developmental component of the lifespan of cortical GABA as measured by MRS of GABA is explored in less depth than in adult or aging cohorts. *Port et al., 2017* and *Silveri et al., 2013* report a maturational increase in GABA+/Cr+PCr in typically developing children and adolescents. The majority of evidence comes from non-MRS work, showing dramatic change in GABAergic function during early life. Human autopsy data describes large changes in both GABA synthesis and receptor

expression (*Pinto et al., 2010*) and animal models describe a shift in GABA from excitatory to inhibitory (*Leonzino et al., 2016*). Yet, in vivo reports in healthy younger populations (particularly infants and young children) are sparse or missing. Several reasons exist for the absence of high-quality MRS data during development with technical challenges being one main challenge. These challenges are not unique to MRS but exist for most – if not all – magnetic resonance modalities. For example, it is well established that imaging of MRS of GABA is highly sensitive to motion (*Edden et al., 2016*; *Mullins et al., 2014*), thus compounding the challenges involved when imaging pediatric cohorts. Many studies in pediatric cohorts, including our own (*Puts et al., 2017*), also suffer from the limitation that age relationships are not reported due to individual studies often studying a restricted age range to *minimize* developmental effects within the cohort.

### Adult

The vast majority of studies in healthy populations focus on age ranges between development and aging to minimize the effect of age on the measures of interest (and for ease of recruitment). However, this limits the reporting of GABA-age relationships within this range. *Mikkelsen et al., 2017* conducted a multisite study collecting GABA+ levels in 272 participants between 18 and 35 years of age, providing a substantial dataset to assess this relationship. The sample size and restriction to healthy adults in this study provide a reasonable representation of normal GABA estimates in the target demographic. Their objective with this study was to report stability of the GABA+ measure across multiple 3 T MRI platforms with systems by GE, Phillips, and Siemens well represented. Voxel placement was selected for the medial parietal lobe. While their original manuscript contains neither a report nor a visualization of the GABA/age relationship, we are able to provide this information for our review (data are freely available from the Big GABA repository, *Mikkelsen et al., 2017*, https://www.nitrc.org/projects/biggaba/). There was no aging-related increase or decrease between age and GABA+/Cr+PCr [$\chi^2(7)$=3.52, $p_{boot}$ = 0.31] in this large cohort of adults between 18 and 35 years of age.

### Aging

Most, if not all, MRS of GABA studies that investigate aging populations report a decrease in cortical GABA as a function of age in both frontal (*Gao et al., 2013*; *Porges et al., 2017a*) and parietal (*Gao et al., 2013*) voxels. *Marenco et al., 2018* also show a decrease in GABA with aging. It is important to note that other reports (*Hermans et al., 2018*; *Maes et al., 2018*) have compared MRS of GABA between defined groups of older and younger adults rather than with continuity across the lifespan. These findings are consistent with continuous approaches, with older adults having reduced GABA. However, a categorical approach comparing two groups does little to elucidate the aging-related trajectory. Manuscripts that employ MEGA-PRESS methodology in a manner that is inconsistent with methods outlined in consensus papers (*Mullins et al., 2014*; *Puts and Edden, 2012*) have insufficient SNR or other technical limitations that have not been considered for this assessment.

In conclusion, an understanding of the link between GABA and age is incredibly important for the study of inhibition across the lifespan, the study of development- and aging-related behavioral and cognitive processes, and the study of health and disease. No study has attempted to study GABA across the entire lifespan. Here, we utilize an IPD-MA approach for combining all existing and eligible cortical edited MRS of GABA data across the lifespan (from childhood development to aging) to build a model that informs us of the best-fit model of GABA across the lifespan. We hypothesize that the model of best fit would be consistent with that of other cortical measures of development and maturation, including indices of white matter (*Lebel et al., 2012*) and gray matter (*Gilmore et al., 2012*; *Gogtay et al., 2004*), as well as EEG power (*Whitford et al., 2007*), revealing non-linear trajectories, increasing during development and slowly decreasing during aging.

## Results

A naïve approach to describing GABA as a function of age is to assume a linear trend. The resulting regression models, fit separately for each of the eight datasets, are depicted in *Figure 1* (scaled with respect to each dataset's geometric mean), with corresponding regression statistics reported in

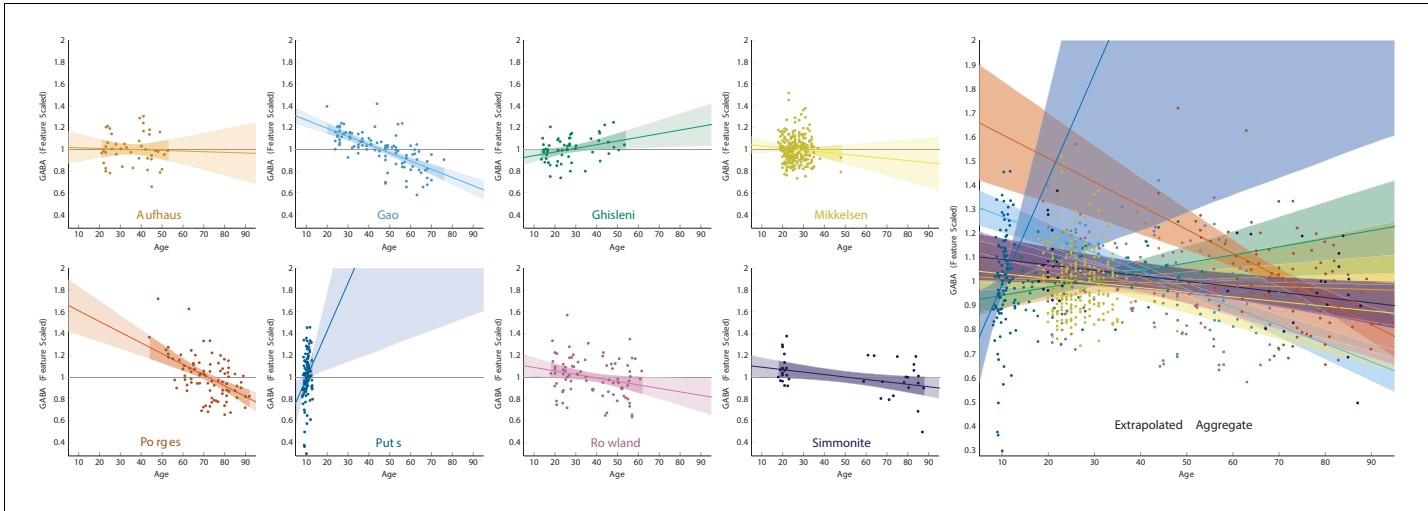

**Figure 1.** Linear relationships between age and γ-aminobutyric acid (GABA) signal, showing that linear extrapolation over the lifespan is not appropriate. In each dataset, GABA was scaled relative to the geometric mean. Linear models were fit for each dataset separately. Dark shaded regions represent the 95% credible interval for the interpolated regression line, given the data from each study and the assumption of a linear effect, whereas the light shaded regions represent the 95% credible interval for the extrapolated regression line.

*Table 1.* A cursory examination of this approach reveals that such an assumption is inappropriate, suggesting that changes in this relationship over time render extrapolation from any of the datasets ineffective. Furthermore, assuming a linear trend is unsuccessful even at describing the data within each study. In six of the eight datasets, the linear trend explained less than 20% of the variance. More dramatically, linear fits were even less successful at predicting extrapolated trend beyond each study's age range, and although the slope in the dataset with the youngest participants was positive, slopes tended to become more negative as the age of the participants increased. When combined into a naïve aggregate of all studies, the extrapolated trendlines show no coherent pattern (*Figure 1*, right panel). Although a linear trend may provide an approximate summary over short periods of time, a linear trend over the entire lifespan is not appropriate. Since unqualified statements about the correlation between age and any neurophysiological measure are, effectively, linear models, *Figure 1* also shows how poorly such statements extrapolate to larger datasets. With this in mind, our meta-analysis did not rely on a linear trend. Instead, we sought to balance our estimation of the overall non-linear function that described the common pattern across datasets with the need to control for systematic differences between studies.

**Table 1.** Regression statistics for simple linear fits.
Intercepts are omitted because rescaling causes them to be entirely determined by the slopes and the mean age. Further study details can be found in *Tables 2* and *3*.

| GABA study | Mean slope | Lower and upper bounds (0.025–0.975 quantiles) | Mean residual $\sigma$ | Lower and upper bounds (0.025–0.975 quantiles) | $R^2$ |
|---|---|---|---|---|---|
| Aufhaus | −0.0008 | −0.0055 to 0.0041 | 0.155 | 0.125 to 0.194 | 0.003 |
| Gao | −0.0075 | −0.0092 to −0.0058 | 0.121 | 0.106 to 0.140 | 0.459 |
| Ghisleni | 0.0034 | 0.0007 to 0.0061 | 0.111 | 0.092 to 0.134 | 0.106 |
| Mikkelsen | −0.0019 | −0.0054 to 0.0016 | 0.129 | 0.117 to 0.142 | 0.005 |
| Porges | −0.0099 | −0.0133 to −0.0064 | 0.172 | 0.148 to 0.200 | 0.279 |
| Puts | 0.0436 | 0.0085 to 0.0795 | 0.215 | 0.188 to 0.249 | 0.058 |
| Rowland | −0.0032 | −0.0060 to −0.0004 | 0.176 | 0.151 to 0.207 | 0.061 |
| Simmonite | −0.0023 | −0.0040 to −0.0005 | 0.160 | 0.127 to 0.203 | 0.156 |

**Table 2.** Neuroimaging acquisition and analysis details for eight studies included in the analysis.

Reference method refers to either reference to water (in estimated concentration/H$_2$O) or as a ratio to creatine plus phosphocreatine (Cr+PCr) and describes whether data was acquired macromolecule-suppressed (γ-aminobutyric acid [GABA]) or as GABA+ macromolecules (GABA+). MRS averages refer to the number of ON + OFF transients. *The manuscript refers to 96 averages. It was clarified with the authors that this referred to 96 ON and 96 OFF averages.

| GABA study | Type of scanner | Analysis software | Reference method | Voxel volume (ml$^3$) | MRS means | TE (ms) | TR (ms) | Voxel location |
|---|---|---|---|---|---|---|---|---|
| *Aufhaus et al., 2013** | 3 T Siemens | jMRUI/ LCModel | GABA/H$_2$O | 24 | 192* | 68 | 3000 | Medial frontal lobe |
| *Gao et al., 2013* | 3 T Philips | jMRUI | GABA+/Cr +PCr | 27 | 320 | 68 | 2000 | Medial frontal lobe |
| *Ghisleni et al., 2015* | 3 T GE | LCModel | GABA+/H$_2$O | 30 | 320 | 68 | 2000 | Left dorsolateral prefrontal lobe |
| *Mikkelsen et al., 2017* | 3 T GE/Philips/ Siemens | Gannet | GABA+/Cr +PCr | 27 | 320 | 68 | 2000 | Medial parietal lobe |
| *Porges et al., 2017a* | 3 T Philips | Gannet | GABA+/H$_2$O | 27 | 320 | 68 | 2000 | Medial frontal lobe |
| *Puts et al., 2017* | 3 T Philips | Gannet | GABA+/H$_2$O | 27 | 320 | 68 | 2000 | Right precentral sulcus |
| *Rowland et al., 2016* | 3 T Philips | Gannet | GABA/H$_2$O | 24 | 256 | 68 | 2000 | Medial frontal lobe |
| *Simmonite et al., 2019* | 3 T Philips | Gannet | GABA+/Cr +PCr | 22.5 | 256 | 68 | 1800 | Medial occipital lobe |

To provide a meta-analytic synthesis of these datasets without biasing our result through an arbitrary choice of our function $g()$, we fit two models: both non-linear and able to accommodate the pattern visible from the individual trendlines.

The estimate for our basis spline model is depicted in *Figure 2* (left). Overall, this time course is characterized by an increase in GABA during childhood, followed by a plateau from adolescence through midlife, and then a gradual decline from approximately 40 years onward. Note that although *Figure 2* (left) depicts the mean spline, any of the wide variety of smooth curves that fit within the shaded credible interval are in principle nominated by this analysis. A future expansion of this model to a larger assemblage of datasets would further refine which of those curves is a suitable candidate for the population's canonical function. The estimate for our log-normal model is depicted in *Figure 2* (right).

To confirm that the models in *Figure 2* did a reasonable job of combining the data despite their differing origins and methodologies, we examined the posterior estimates of the scaling factors. *Figure 3* (left) shows the posterior estimates for each dataset's scaling factor for our basis spline model, whereas *Figure 3* (right) shows the estimates for the log-normal model. These scaling factors were

**Table 3.** Descriptive statistics for eight studies included in the analysis.

This gives a basic description of sample size and age range for the eight datasets. Additionally, *Figure 6* depicts the distribution of ages using a raincloud plot (*Allen et al., 2018*).

| GABA study | # of subjects | Mean age | Age (SD) | Age range | Reference |
|---|---|---|---|---|---|
| Aufhaus | 44 | 35.5 | 10 | 21–53 | *Aufhaus et al., 2013* |
| Gao | 96 | 45.7 | 14.5 | 20–76 | *Gao et al., 2013* |
| Ghisleni | 55 | 27.2 | 11 | 13–53 | *Ghisleni et al., 2015* |
| Mikkelsen | 220 | 26.5 | 4.9 | 18–48 | *Mikkelsen et al., 2017* |
| Porges | 86 | 71.8 | 10.6 | 43–92 | *Porges et al., 2017a* |
| Puts | 101 | 10.3 | 1.2 | 8–13 | *Puts et al., 2017* |
| Rowland | 82 | 38.0 | 13.7 | 18–62 | *Rowland et al., 2016* |
| Simmonite | 38 | 50.1 | 29.2 | 18–87 | *Simmonite et al., 2019* |

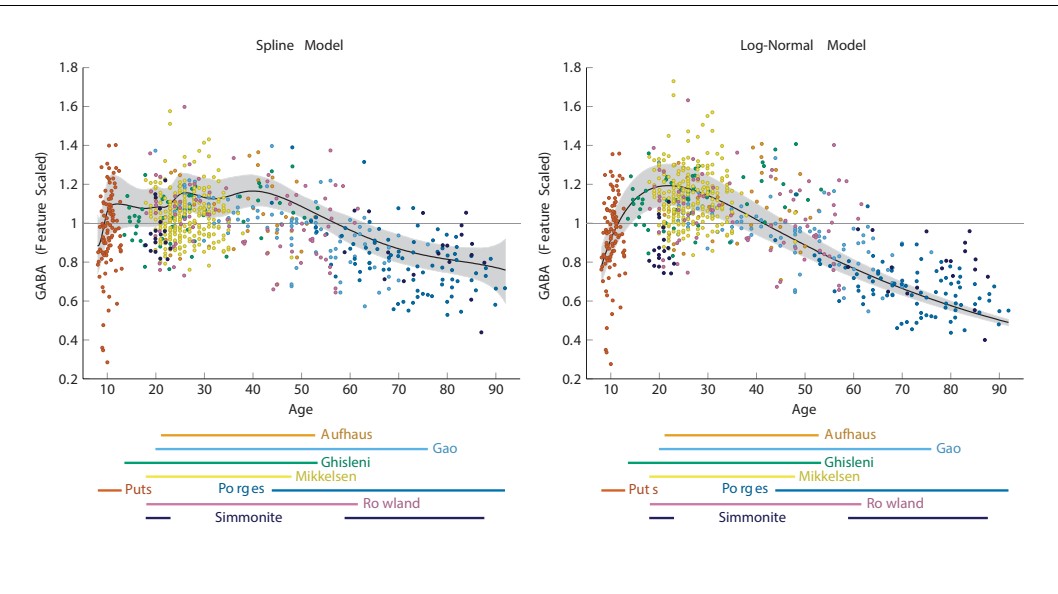

**Figure 2.** Non-linear regression models of γ-aminobutyric acid (GABA) signal integrating all data simultaneously. The shaded region depicts the 95% credible interval for the mean. (Left) Penalized basis spline model. (Right) Log-normal model.

given broad prior distributions. Rather than trying to characterize possible contributions to each scaling factor (e.g. by reference method or scanner manufacturer), we allowed the scaling factors to be estimated in a manner that was agnostic to any label beyond which dataset each observation came

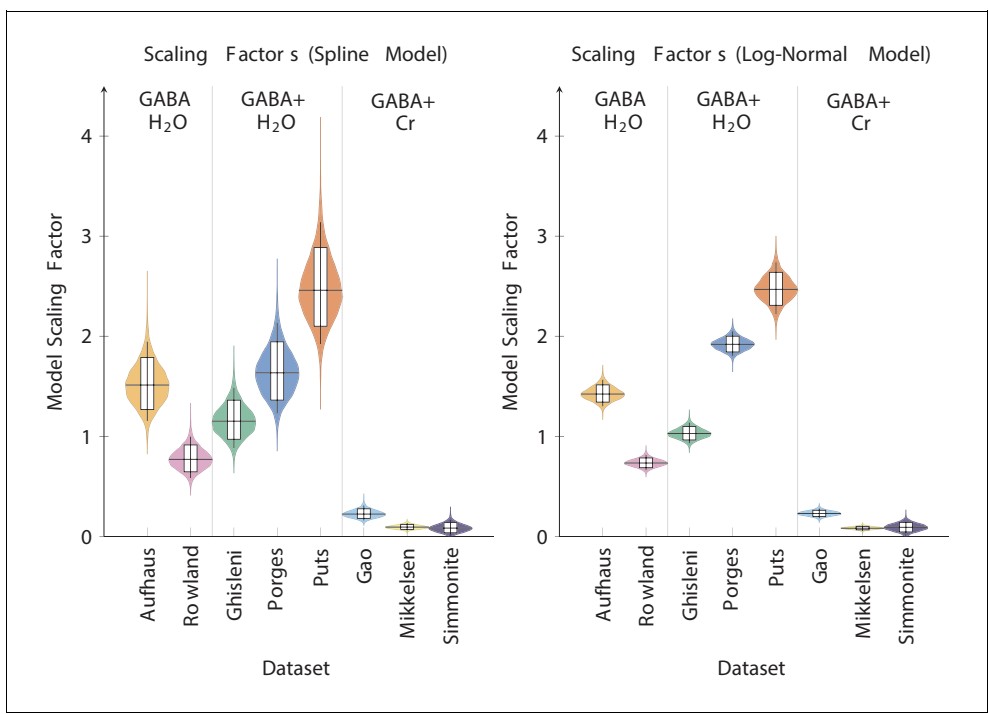

**Figure 3.** Posterior estimates of the relative scaling factor $F_s$ for each study in the penalized basis spline model (left) and the log-normal model (right), sorted by reference method. Boxes represent the 80% credible interval for the posterior estimate, whereas whiskers represent the 95% credible interval.

from. Thus, scaling factors were governed only by (1) the observed values of the data and (2) the constraints of the model.

Despite our decision to keep our priors broad and agnostic, the model nevertheless recapitulates expected differences in measured outcomes. The Big GABA series of meta-analyses (*Mikkelsen et al., 2017*; *Mikkelsen et al., 2019*) are the largest published multi-site studies to date characterizing GABA measurement, and our estimated scaling factors line up with the score ranges consistent with those reported in that series of studies. For example, our three Cr+PCr-referenced datasets (Gao, Mikkelsen, and Simmonite) all yielded scaling factors well below 1.0, consistent with the measurement scale typically observed in GABA+/Cr+PCr studies (*Mikkelsen et al., 2017*), whereas water-referenced data yielded scaling factors one or two orders of magnitude larger than Cr+PCr-referenced data, also consistent with typical measurement (*Mikkelsen et al., 2019*). In this respect, our estimated scaling factor not only put our observations on a comparable scale but did so in a way that provides a sanity check against published findings.

Importantly, however, the scaling factor estimates are themselves distributions, and those distributions depend not only on the variation in the original data but on their covariance with other variables and on the model assumptions. The scaling factor estimates display less uncertainty in the log-normal case because that model imposes much stronger assumptions on the time course of the data than would be expected from a spline model. The estimated scaling factor for the Mikkelsen data varies less than that of the Simmonite data in part because the former dataset is nearly six times as large. Since every parameter estimate within each model covaries with every other parameter

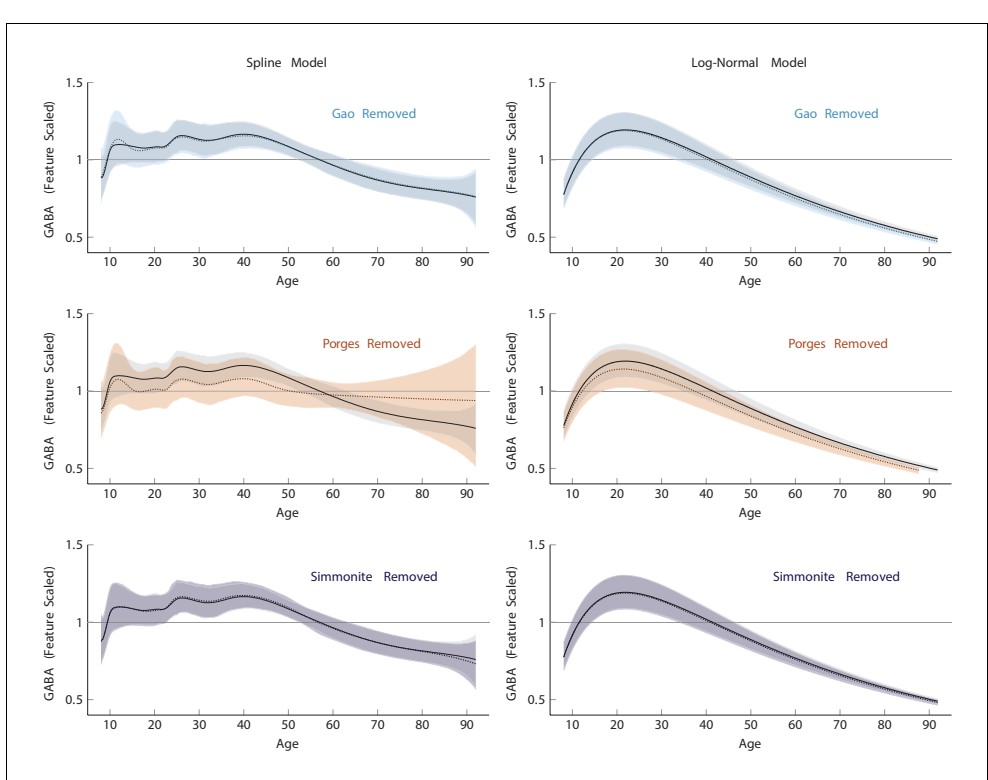

**Figure 4.** Non-linear regression models of γ-aminobutyric acid (GABA) signal performed using a leave-one-out (LOO) cross-validation approach. The gray shaded region and black line depict the 95% credible interval for the mean for the model and the colored shaded region and dotted line show the model with the respective data left out. In all cases, a similar overall trajectory to the full data is implied by each of the subsets, albeit with greater variation in the posterior estimates.

The online version of this article includes the following figure supplement(s) for figure 4:

**Figure supplement 1.** To accompany *Figure 4*, we performed additional analysis on frontal data only (removing Mikkelsen et al. and Simmonite et al., respectively) to investigate whether the inclusion of non-frontal regions biased our data.

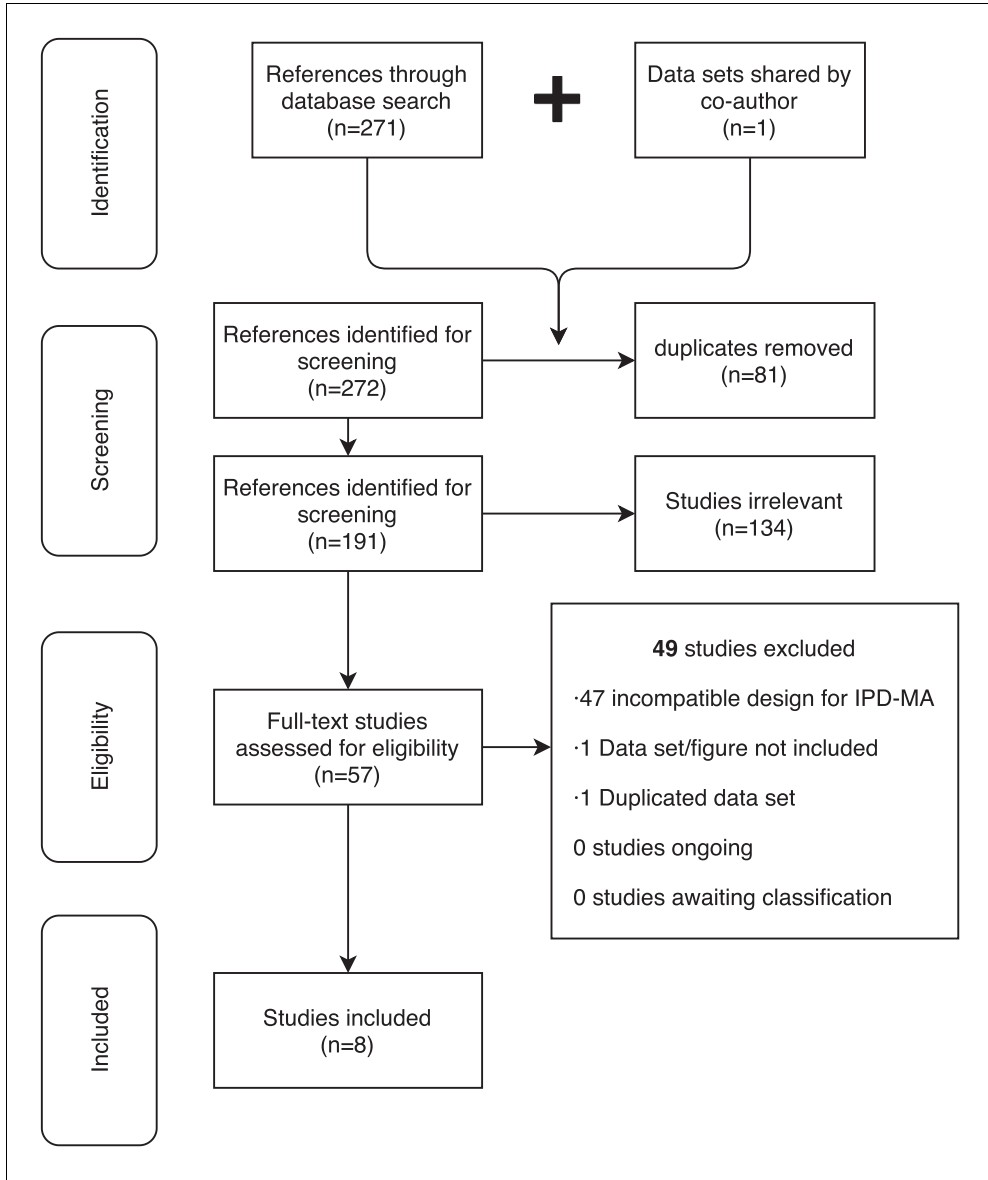

**Figure 5.** PRISMA 2009 flow diagram of study identification and inclusion.

estimate, and furthermore is relative to the other reported factors, it is important to treat these scaling factors as strictly relative and contingent on the current data and model assumptions.

To ensure that the directionality of the 'late-age' component was not driven by differences in Cr +PCr, we performed an additional 'leave-one-out' (LOO) cross-validation approach (*Arlot and Celisse, 2010*) for each of the three datasets that included late-age participants. *Figure 4* shows that when datasets are left out, the general shape of both models remains the same, with wider 95% credible intervals for the oldest of age due to limited available data. This suggests that the overall pattern of a negative slope in late age is unlikely to be driven by Cr+PCr differences, as the data contributed by *Porges et al., 2017a* made the largest contribution to this slope (having the most observations to contribute) and relied on a water-based reference method. That said, since the Porges dataset contains most of the observations over 60 in this analysis, its removal leaves only a handful of cases from the Simmonite dataset as evidence for late-life model estimates, resulting in substantial uncertainty in the spline model. By contrast, the age ranges represented by Gao and Simmonite have good coverage by other studies, so their removal does not have as large an impact on the estimate.

*Figure 4* also helps to demonstrate the importance of model assumptions: The removal of the Porges dataset affords the spline model the possibility that the model of best fit might level off or even become positive in late life, whereas the log-normal model's rigid shape is only able to predict a downward trajectory. On the one hand, the log-normal model can probably be criticized for its inflexibility in this respect; however, on the other, this demonstrates that without much data to guide it, the spline model is perhaps too agnostic about the direction of the effect. The spline model's advantage is that it can be persuaded to turn, provided the evidence is consistent with that conclusion. The log-normal, by contrast, is too inflexible to accommodate a change in direction in late life regardless of the volume of data it is provided.

We additionally also explored the potential effect of region by performing the analysis on only frontal data (removing Mikkelsen et al. and Simmonite et al., respectively) with no substantial impact on the direction of the slope over time. This information can be found in *Figure 4—figure supplement 1*.

## Discussion

Here, we show that the aging-related trajectory of GABA across the lifespan is characterized by a fairly quick increase in GABA during development, followed by a flattening during adolescence and by a subsequent slow decrease with aging. It should be noted that we included studies reporting GABA levels in institutional units and as measured ratios. No single study with a limited age range reveals such a relationship (as evidenced by *Figure 1*) – it is only by meta-analysis of multiple datasets that we are able to identify this relationship. Here, we will discuss the case for a biological mechanism that drives a non-linear trajectory of GABA measures with age and then discuss methodological and biological factors that may influence these results. Finally, we review the implications of our evaluation and suggest potential directions of future research characterizing the trajectories of GABA over the lifetime.

### Biological mechanisms

Although previous work, including the authors' own, has generally reported a linear relationship between GABA and age within a given stage of life, it is far more plausible that differences across the lifespan are non-linear, in line with other biological effects. For example, while *Gao et al., 2013* might capture a decrease in GABA with age, the broad age range studied in *Ghisleni et al., 2015* might prevent observation of a linear relationship. Indeed, a decline in GABAergic interneurons with age has been widely reported in animal models (*Hua et al., 2008*; *Stanley et al., 2012*). Post-mortem data from human samples showed a reduction in the 65 isoform of GAD (the enzyme responsible for the production of GABA, GAD) in visual cortex, suggesting reduced GABA production with aging (*Pinto et al., 2010*). MRS of GABA by itself cannot provide sufficient resolution to determine what these reductions in GABA reflect; we can only make vague assumptions and conclusions on the relationship between brain structure, cognitive function, and potential molecular mechanisms.

In considering a non-linear model for GABA across the human lifespan, we find three major stages: (1) a developmental stage where GABA measures increase, (2) a stabilization phase during adulthood where GABA concentrations remain mostly stable, and (3) a gradual descending period of GABA with advanced aging. This is consistent with previous studies of brain structural and cognitive function across the lifespan. Non-linear trajectories have been reported in aging effects of total gray matter (*Lenroot et al., 2007*; *Sussman et al., 2016*) and cortical thickness (*Shaw et al., 2008*). Diffusion tensor imaging studies show non-linear aging-related differences through childhood and adolescence (*Lebel and Beaulieu, 2011*), with an 'inverted U-shaped' trajectory that peaks at approximately 40 years of age (*Bendlin et al., 2010*; *Good et al., 2001*; *Lebel et al., 2012*; *Westlye et al., 2010*). A similar trajectory pattern for cognitive abilities is reported in memory, verbal ability, and inductive reasoning (*Kobayashi et al., 2015*), as well as word recall, verbal fluency, math skills (*Whitley et al., 2016*), and behavioral inhibition (*Williams et al., 1999*).

### GABA differences throughout development

An increase in cortical GABA could be extrapolated from the proliferation of new GABAergic neurons during development. GABA is thought to be linked to the myelination of frontal white matter trajectories by controlling oligodendrocyte precursor cell activity through the developmental phase

(*Ghisleni et al., 2015*; *Vélez-Fort et al., 2012*). An increase in cortical GABA concentrations could also be the result of increased synaptic activity of GABAergic neurons during development. This potentiation-like mechanism of cortical GABA is supported by studies where astrocytes have been shown to mediate plasticity of rodent hippocampus (*Kang et al., 1998*) and visual cortex (*Chen et al., 2012*), suggesting a possible potentiation-like mechanism of cortical GABA concentrations. In addition, upregulation of GAD could lead to increased production of GABA and indeed, GABA levels as measured with MRS have been shown to relate to expression of the GAD1 gene (*Marenco et al., 2010*). GABAergic neuronal function is reported to become more efficient through synaptic pruning and long-term depression during development (*Paolicelli et al., 2011*; *Wagner and Alger, 1995*; *Wu et al., 2012*). It should be pointed out that no eligible MRS studies of GABA were available for infancy and early development, which is a significant gap in the literature that should be addressed in future work.

## GABA differences throughout aging

Decreased GABA concentrations during aging are most likely linked to gray matter atrophy and demyelination associated with pathology and normal aging. Animal models have explored the physiological underpinnings and functional implications of these changes. Rodent studies have shown a reduction in the number of interneurons expressing GAD in the medial prefrontal cortex. These reductions were accompanied by altered spatial working memory, linking altered GABA function to altered behavior in aging (*Spiegel et al., 2013*). Additionally, rodent models of cognitive decline have shown altered function at both GABA-A and GABA-B receptors (*McQuail et al., 2015*), as well as a reduction in GABAergic neurons in cats and monkeys (*Hua et al., 2008*; *Leventhal et al., 2003*).

## GABA differences across the lifespan

Very few non-MRS studies have assessed changes or differences in GABA function across the lifespan. In those that report differences in the GABAergic system across the lifespan, a pattern of change similar to our findings has been presented. Using GAD labeling methodology in the human visual cortex, GAD65 has been reported to increase early in life and gradually decrease during aging (*Pinto et al., 2010*). Provocatively, they show GAD67 to be stable across the lifespan. Similarly, a recent cross-sectional exploration of sex differences in young and old adults found reduction in GAD65, but not GAD67 in the superior temporal gyrus of females (but not males). No change in GAD65 or GAD67 was found in other regions, though prefrontal cortex was not reported (*Pandya et al., 2019*). We are unaware of other reports of normal human aging-related differences in GAD; however medial temporal lobe reductions (prefrontal cortex was not reported) in GAD65, but not GAD67, have also been reported in Alzheimer's disease (*Schwab et al., 2013*). Given the more specific relationship between GAD65 and neurotransmission, this may underlie reports of GABA-associated alterations in cognitive function that occur during periods of GABAergic change (*Porges et al., 2017a*) throughout the lifespan. Future studies should aim to address the relationship between these two different functional isoforms, brain GABA levels, and function in both health and disease.

## Regional differences

It can be assumed that regional differences in GABA are not homogeneous. We do not presume that the model we have presented is characteristic of aging-related differences in GABA in all neural tissues due to well-known ontological and aging-related regional variation in tissue that influences GABA (*Lebel et al., 2012*). One could even argue that regional differences are likely to exist within smaller regions of the frontal lobe. However, limiting this regional selectivity even further would have not allowed for our analysis, as (1) we would not be able to correct for covariation between frontal and parietal regions, and (2) inclusion of multiple observations per participant would violate the independence of individual data points in our analysis. Because all data we included from a particular study were restricted to the same cortical region, differences between regions would be absorbed by the studies' scaling factors $F_s$ and would be indistinguishable from methodological systematics (further discussed below). We therefore decided to include frontal data when available, and other regions when not available, but given the potential regional differences, we also show our

aging-related trajectory without the inclusion of *Mikkelsen et al., 2017* and show that the model shows a similar trajectory for frontal-only data (*Figure 4—figure supplement 1*). The majority of published studies on MRS of GABA focus on cortical, rather than subcortical, regions of interest. The few publications that describe subcortical GABA often show trends that are difficult to associate with cortical GABA levels. For instance, GABA was negatively associated with age in subcortical voxels but positively associated with age in anterior and posterior cortical voxels (*Ghisleni et al., 2015*). Subcortical GABA undoubtedly plays an important role in brain function, as evidenced by increased GABA concentrations in subcortical basal ganglia that have been associated with schizophrenia and depression (*Puts and Edden, 2012*). However, the role of subcortical GABA—measured via MRS—in cognitive function is not thoroughly investigated in the literature, and thus there are too few studies to perform a suitable meta-analysis. In the current review, we limit our discussion to GABA concentrations in cortical voxels in order to provide a meaningful initial evaluation of GABA over the lifespan. Both the publication of future participant-level data and the release of participant-level data from past studies will increase the sample available to the field, which in turn will make possible the rigorous examination of the contributions of these covariates.

## Methodological differences

Inconsistencies in aging-related differences in GABA estimates may stem from differences between study methodologies or inherent structural differences across the lifespan. We discuss each of these in turn below.

Quantification of MRS is relative and expresses the ratio between the signal of interest and an internal reference signal. The most widely used references for GABA measures are the Cr+PCr signal in editing-off spectra and the unsuppressed water signal from the same volume (*Alger, 2010*; *Mullins et al., 2014*). Each quantification approach has its advantages and disadvantages. For instance, the Cr+PCr at 3.05 ppm has minimal chemical shift from the GABA signal at 3 ppm (*Mullins et al., 2014*) and is acquired during the MEGA-PRESS sequence. Therefore, its signal comes from the same location as the GABA signal, and it does not require a separate acquisition. In contrast, the water signal represents a more concentrated chemical yielding a higher SNR, but it may also introduce error in estimates of location due to chemical shift effects (*Mullins et al., 2014*) when not acquired from the same voxel as GABA and on some scanners requires a separate acquisition (see Choi et al. for a discussion). Furthermore, the Cr+PCr signal arises only from tissue, whereas water signal arises from tissue and cerebrospinal fluid (CSF) with substantially different relaxation behavior. A small number of studies have looked at the relationship between Cr+PCr and age (*Ding et al., 2016*; *Lind et al., 2020*) showing an increase or no change with age (for a review, see *Cleeland et al., 2019*). The impact of increasing Cr+PCr during aging would be to make the any aging-related decrease in GABA/Cr+PCr more pronounced, and the potential of this to contribution should not be ruled out. However, Gao et al. investigated both GABA/Cr+PCr and GABA/NAA and found similar aging-related reductions in GABA referenced to both molecules. Given that NAA has, in contrast to the increase of Cr+PCr, been shown to consistently decrease in aging (*Cleeland et al., 2019*), it makes it unlikely that Gao's aging-related reduction was solely driven by a Cr+PCr increase during aging. Still the potential for a Cr-bias on the final model that needs to be considered. Our LOO approach to the 'late-age' data (*Figure 4*) found no substantial differences in the final models, providing some support to the notion that differences in Cr+PCr are not driving the GABA findings presented here, but the impact of reference compound needs to be considered.

One of the cornerstone assumptions of our analysis is that, although the use of different references results in different fundamental units, participants within any given study using a given method can be compared to one another in relative terms. If, for example, participants in the oldest quartile of a study show only 80% of the GABA signal on average of participants in the youngest quartile, then there is a basis for reporting a reduction in GABA regardless of the reference method used. In other words, our analysis presumes that the measures' scores in any given study share a common scaling factor (which we denote as $F_s$) that can be estimated in order to factor out its influence, thus allowing studies to be compared in terms of their relative expressions of GABA. This approach thus relies on different reference methods displaying consistent sensitivity within each study; that is, participants with comparatively low scores in a study should be measured with a precision that is similar to that of participants with comparatively high scores. If this assumption is violated, a more complex analysis would be needed to address it, such that each participant's contribution is given a weight as

a function of their measurement error relative to other participants in the same study. Provided this assumption is met, however, simultaneous estimation of between-list scaling factors and within-list participant differences provides a robust means of integrating multiple studies into a single overall trajectory.

Another limitation is that voxel location can be inconsistent between studies (e.g. the medial pre-frontal cortex in one study may not be localized the same way as in another study; also see a recent review: *Peek et al., 2020*). This becomes more problematic in younger cohorts where, due to smaller intracranial volumes, the methodologically limited size of the voxel (*Mullins et al., 2014*) necessarily incorporates a proportionally larger fraction of the brain.

It is well known that voxel tissue composition has a significant impact on the quantification of GABA levels (*Harris et al., 2015a*). In many cases, tissue correction is appropriate due to existing partial volume effects (*Barker et al., 1993*; *Christiansen et al., 1993*; *Danielsen and Henriksen, 1994*; *Ernst et al., 1993*; *Hennig et al., 1992*; *Kreis et al., 1993*; *Thulborn and Ackerman, 1983*). Researchers frequently tissue-correct GABA values by segmenting the T1 weighted structural images (*Ghisleni et al., 2015*). Interestingly, none of the studies included in this review showed significant differences in segmented tissue content with age. This is surprising because other studies

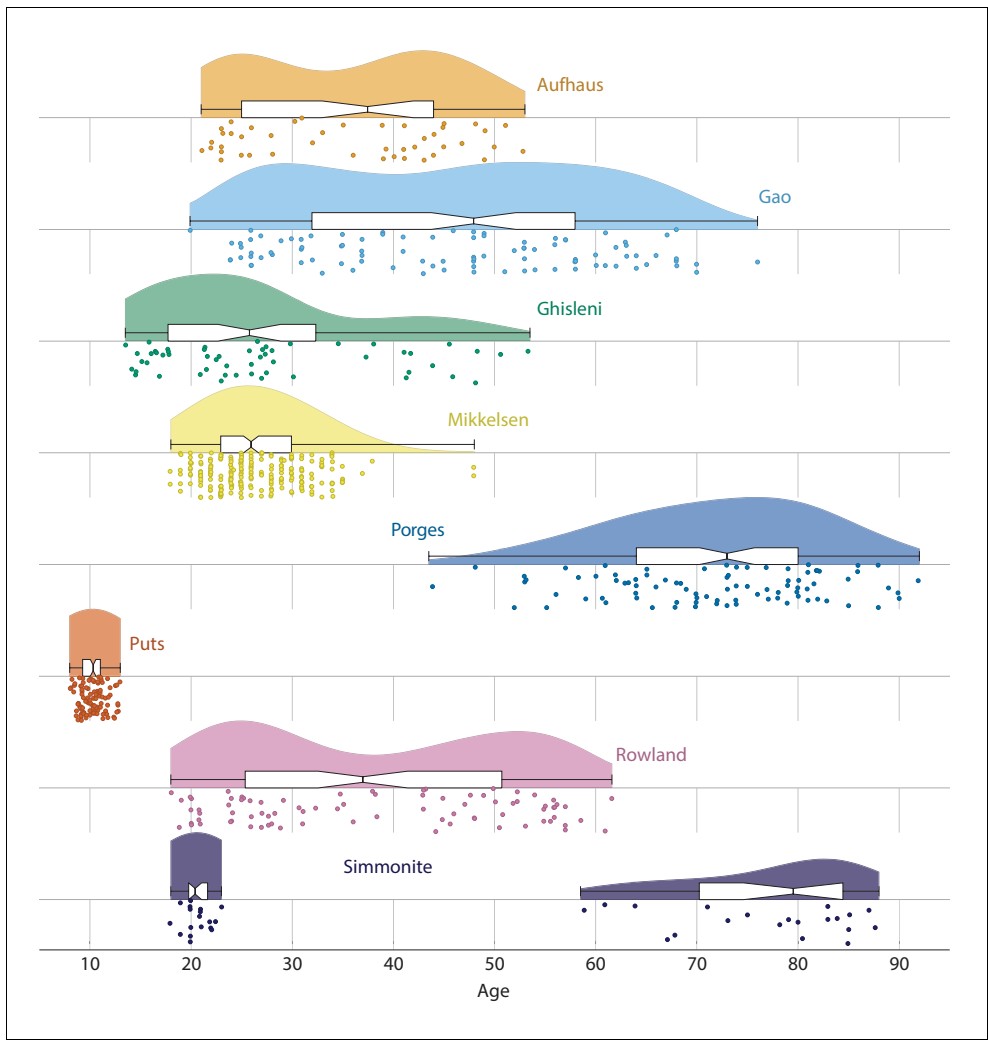

**Figure 6.** Raincloud plot depicting the age distribution in each of the eight included datasets. Plotted densities are scaled within study. Boxes represent the first and third quartiles, while whiskers represent the range of the data. The central notch corresponds to the bootstrapped confidence interval for median. Kernel density estimates for each dataset were computed using a Gaussian kernel according to *Silverman, 1986*. The relevant analysis script for these densities is included as supplementary material.

(*Maes et al., 2018*; *Porges et al., 2017b*) showed that aging-related differences in GABA depend on tissue correction due to atrophy being common in older participants. Unfortunately, there is no consensus on the appropriate tissue correction approach except that accounting for different compartments when using a water reference is strongly advised. The potential effect of tissue composition on GABA levels is limited if a study focused on a young cohort where aging-related atrophy would be negligible. All of the studies included in this review assume a linear model to represent change in GABA. Thus, while tissue correction could potentially contribute to the variance and differences between MRS-GABA studies, they are unlikely to explain the discrepancies between study populations of different age ranges.

In water-referenced studies of aging-related differences in older cohorts (where bulk tissue changes are most likely to impact this relationship), accounting for CSF in the tissue correction approach does not remove a significant relationship between age and cortical GABA (*Porges et al., 2017b*). Given our concern was with relative differences in GABA, our analysis approach used a feature scaling approach to examine within-study variation as an estimation problem that was simultaneous with the estimation of the time course. Given this approach, the scaling factor $F_s$ for each study $s$ is conceptually very similar to a method of correcting 'house effects' in the analysis of political polling data (*Jackman, 2005*), allowing pollsters that show systematic bias (e.g. to a particular party) to be included in aggregate measures of public opinion. Furthermore, a limited effect of tissue is also suggested by consistency between findings of GABA+/Cr+PCr and estimated concentrations. There is no significant difference in the correlation for GABA × age in the overlapping age range of 43–76 years between Porges et al. (GABA+/$H_2O$) and Gao et al. (GABA+/Cr+PCr) who looked at comparable samples (Fisher R-to-Z [$z = 1.48$, $p = 0.19$]) showing consistency between studies focusing on the same population (see also the discussion above).

As for other methodological differences (*Table 2* and *Table 3*), studies used a variety of scanner vendors: a variation that is known to contribute to between-site variability (*Mikkelsen et al., 2017*) but is unlikely to lead to substantial difference in the age relationship. Furthermore, a variety of analysis techniques have been used, making direct comparison between studies problematic. A given analysis pipeline is unlikely to be biased to substantial differences in the age relationship. By rescaling the data in a way that emphasized scale ratios within each dataset, we minimized the impact of differences in site and vendor-driven GABA magnitude estimations. Furthermore, inconsistencies between reference compound (e.g. water or Cr+PCr) additionally complicate interpretation of the findings. Future studies should provide concentration values across the lifespan; we recommend reporting of values both as metabolite and water ratios to look at within-study consistency and robustness of the effect of GABA. Recent developments in MRS of GABA methodology will allow for this, even when data is collected on multiple scanner platforms (*Saleh et al., 2019*).

It should be noted that all datasets used for this meta-analysis were cross-sectional. Although it is tempting to draw longitudinal inferences from our analysis, there are several limitations in doing so. For example, the present data provide no insight into long-term survival trends, so the population that is represented at age 20 likely differs in various ways from the population that is represented at age 70 and we cannot rule out a relationship between GABA and survival-related confounds. Additionally, because our method simultaneously estimated the scaling factor and the time course for each dataset, areas of minimal overlap between datasets (e.g. at around 13 years) are particularly uncertain.

While here we discuss potential methodological origins of bias, it should be noted that many of these cannot be extracted from the literature. We opted to use the MRS-Q as a guideline to assess quality, and recent consensus (*Lin et al., 2021*) also addresses issues in reporting important data acquisition, analysis, and quality information moving forward, including measures that may impact quantification, including proxy measures of motion, fit errors/CRLB, and shim.

The present analysis provides a model of the lifespan based on the data that is currently available. It would be valuable to test these in a longitudinal study of MRS of GABA across the lifespan. Even if it is not feasible to measure GABA in the same individual over a 70-year span, obtaining multiple estimates per participant over a moderate length of time (e.g. 5 years) would greatly facilitate estimating rate of change over time.

## Conclusion and future research

This review considered research of cortical GABA concentrations over the lifespan. After combining datasets, we conclude that a linear model of GABA over the lifetime is not supported. Instead, consistent with other developmental aging studies of neurophysiology and cognitive function, we propose non-linear models to describe lifespan GABA measurements. A log-normal trajectory provides a satisfactory parametric description of the life course for the time being, but large and longitudinal datasets may necessitate the use of nonparametric regression strategies to best characterize the age-GABA relationship.

In the future, it will be important to investigate the neurophysiological and anatomical processes that drive apparent differences in bulk metabolite and neurotransmitter levels. While it is clear that GABA changes with age and seems to follow trends reported in other lifespan datasets, non-linear relationships in GABA and other neurotransmitter and neurometabolite (e.g. NAA, myoinositol, and choline) concentrations merit further exploration. Glutamate in particular, given its close functional and metabolic relationship with GABA, would warrant a similar lifespan approach. In fact, Glx (combined glutamate and glutamine) can typically be quantified from the edited difference spectrum as a result of MEGA-PRESS but most studies do not report Glx. This may be due to the absence of clear hypotheses regarding Glx and a mechanistic focus on GABA, or simple omission of these data. Furthermore, it would be important to link these findings to anatomical measures underlying the different neurochemical measures, such as evidence of non-linear cortical thinning and histological studies of GABAergic neurons (*Pandya et al., 2019*; *Vidal-Pineiro et al., 2020*).

Future inquiries would benefit from recruiting cohorts that encompass the entire lifespan, and sufficient representation and characterization of males and females, as sex differences have been reported in both GABA (*O'Gorman et al., 2011*) and GAD65 (*Pandya et al., 2019*). This would address that a limitation of the present study was the inability to distinguish males from females in the figures data was extracted from. Studies of development including those of infants and young children are virtually non-existent but are crucial given the importance of GABA in early development. Furthermore, alterations in GABA have been seen in neurodevelopmental and neurodegenerative disorders—a better understanding of abnormal GABAergic function early in development may elucidate this relationship, point to potential early-intervention targets, or explain variability in response to pharmacological treatments for these conditions.

Here, we report on the relationship between GABA and age across the lifespan. It is well known that GABA contributes to cognition and perception, which in and of themselves change with age. It would be extremely interesting to apply an IPD-MA approach to the investigation of how aging-related differences in GABA might correlate with cognition. However, this is greatly complicated by the wide variety of cognitive measures used across studies, which would be much more difficult to standardize across studies than feature-scaled GABA as measured using an MRI machine. While pursuit of this work is notably challenging, conducting such research is undeniably crucial going forward.

While many papers exist that report on aging-related differences that are consistent with our asymmetrical lifespan trajectory, these are often reported as group or cohort differences without the presentation of individual data points necessary for such meta-analyses as applied here (*Hermans et al., 2018*; *Port et al., 2017*). Collaborative and group science approaches are becoming increasingly important in generating large datasets that allow for a broader and larger-scale application of this work and we hope that sharing data, or at least reporting individual data points, becomes more common in the future even in studies where age may not be the main focus.

## Materials and methods

We conducted and reported this systematic review in accordance with the PRISMA statement (*Moher et al., 2009*) and we used an IPD-MA approach (*Debray et al., 2015*).

A systematic literature search was performed in two iterations by BF and EP to retrieve studies in which MRS of GABA using the MEGA-PRESS method was collected in the human brain from voxels that included the frontal lobe. In the first iteration, a search was performed using Google Scholar and Medline with the following combination of terms: (GABA OR GABA+ OR γ-aminobutyric acid OR gamma-Aminobutyric acid) AND (MRS OR Magnetic Resonance Spectroscopy) AND (MEGA-PRESS OR MEGA-PRESS OR MEshcher-GArwood Point RESolved Spectroscopy OR edited). Both

GABA+ and macromolecule-suppressed measures were deemed inclusive. We purposely did not include unedited measures of GABA as it is not clear whether these allow for specific and reliable estimation of GABA as per consensus (*Choi et al., 2021*; *Mullins et al., 2014*). The following constraints were applied to limit results: the result should be (1) a full-text article or a conference abstract, (2) peer-reviewed, (3) written in English, (4) included in the publication must be a scatter plot with GABA by age suitable format for extraction of individual human subject data via WebPlot-Digitizer. The search was conducted on April 2, 2019, resulting in a total of 271 studies. Out of these, 55 were relevant.

## Additional criteria

### Study design

A second step was performed by BF, EP, and NP to exclude based on the following criteria: studies must report GABA, acquired using MEGA-PRESS, in at least one cortical voxel (subcortical voxels were excluded) in populations reported as being 'healthy', 'normal', or free of reported relevant disorders. To exclude a potential region effect, when a frontal voxel (any inclusion of frontal cortex) was available, we included that voxel. We chose to focus on frontal regions given this region is most-widely studied with respect to aging-related differences (see Introduction), making it suitable for an individual data meta-analysis; we are first and foremost interested in its relation to behavioral and cognitive associations with GABA. Furthermore, we can link neurochemical aging-related differences to other established brain indices in frontal regions during aging (e.g. white matter). Additionally, we were interested in the apparent contradictory reports in the change trajectory in frontal regions which we attempt to reconcile and finally, frontal regions allowed us to use data-informed predictions on aging-related differences in GABA to support the design and interpretation of future work.

However, to maximize the available data, if no frontal regions were reported, we used the other cortical voxel only as was the case for Mikkelsen et al. If a study sampled multiple voxels, we only extracted data from a frontal voxel to prevent multiple sampling of a single subject as this would include in inclusion of non-independent datasets. If multiple frontal voxels were available, we chose the dataset with the largest sample. Then, data were only included when they originated from published figures of sufficient quality for data extraction of individual data points, and psychiatric or neurologic populations, if presented in the same figure could be distinguished. Finally, the data extracted required contiguous age ranges of 5 or more years for inclusion in the analysis. Studies failing to satisfy all criteria were deemed incompatible with the IPD-MA approach (as described below). Duplicate datasets were excluded as well.

### Data quality

A third step was performed to assess whether studies adhered to consensus quality assurance criteria for data collection, analysis, and reporting. For this purpose, two coauthors (NP and EP) evaluated all remaining studies using the MRS-Q, which was specifically designed for MRS and based upon consensus documentation (*Peek et al., 2020*). MRS-Q identifies a limited number of edited MRS approaches to report and is specific for edited MRS and consistent with the recently published MRS standards in reporting (*Lin et al., 2021*). These criteria are necessary methodological minimum requirements (e.g. voxel size) as reported in existing consensus work for GABA quantification (*Mullins et al., 2014*; *Wilson et al., 2019*; *Peek et al., 2020*). Spectral quality was not a criterion for exclusion, as individual data spectra or quality metrics (e.g. linewidth, CRLB, etc.) are rarely if ever published in manuscripts used for extraction in our IPD-MA approach.

Of the 55 relevant studies retrieved in the systematic search, only seven included figures suitable for data extraction or had data freely available in available online repositories. One dataset was only partly published (*Puts et al., 2017*) but was supplied in full by authors of this manuscript NP and RAEE, for a total of eight datasets. While full-text article and conference abstracts were included in the initial search, no conference abstracts meet our inclusion criteria, thus all included datasets were from peer-reviewed manuscript. At final review, these datasets were reviewed for consistency in research methods (see *Figure 5* and *Table 2*), evaluated for age distributions (see *Figure 6* and *Table 3*), and combined in aggregate (see *Figure 2* and *Table 1*).

## Risk of bias

All eight studies included in the systematic review included both male and female participants, although the pediatric data were somewhat skewed to include more males. No data were excluded based on acquisition parameters as assessed with the MRS-Q. Most studies were well matched for acquisition and followed published recommendations (*Mullins et al., 2014*). No exclusions were made based on the direction of the correlations. With the exception of two publications (*Gao et al., 2013*; *Porges et al., 2017a*), the age-by-GABA relationship was not a primary outcome and was therefore unlikely to have been a driver of publication bias in the majority of studies included. However, a risk of publication bias cannot be ruled out.

Individual data points were extracted from figures using WebPlotDigitizer (*Rohatgi, 2019*). None of the data included were corrected for voxel tissue fractions (see Discussion). MEGA-PRESS sequences can vary between and within MRI vendors; these can impact editing efficiency and in turn absolute quantification (*Harris et al., 2015b*; *Saleh et al., 2019*). However, the consequence of this will not impact the within-site relationship to age (the metric used in this review) as the consequences of such variation are stable within-site and function as a scaling factor (*Mikkelsen et al., 2018*).

## Statistical methodology

Our meta-analysis made use of an IPD-MA approach (*Debray et al., 2015*). The chief advantage of this approach is that it allows the analyst to account for the evidence provided by the individual observations recorded in each study while also accounting for any systematic differences between studies using the framework of a multi-level model that evaluates both the data overall and within the context of each separate study as a simultaneous estimation problem. As a result, methodological steps like estimating an overall 'weight' associated with each study in order to determine how to combine reported statistics is rendered unnecessary, as the relative effect size and uncertainty of each study is communicated to the model by the data themselves (*Riley et al., 2010*). Such an approach is especially important in non-linear regression paradigms because the goodness of fit provided by each value of a parameter is interdependent with every other parameter. Put another way, optimal parameter values cannot be estimated for each parameter in isolation of the others, as the posterior distributions for the parameters covary with one another. As such, the meta-analytic models' uncertainty for any value of a continuous predictor depends on the complex covariance of parameters. In the present analysis, we are concerned with scaling every study relative to every other study, so it would be inappropriate to convert the data to some kind of standard score prior to combining them; instead, this standardization should happen simultaneously with the estimation of other model parameters.

Our general statistical framework in this analysis was to assume that some unknown 'canonical function' describes the average change in the feature-scaled GABA signal over the lifespan as a function of age. In other words, given participant $i$ in study $s$, their age is denoted by $x_{s,i}$ and the relative change in GABA over the lifespan is given by the function $g(x_{s,i})$. This change cannot be directly observed though imaging, but is instead inferred indirectly from a measurable reference (in our case, either water or Cr+PCr). As such, the mean observed effect in each study has an unknown feature scaling factor $F_s$, which is expected to vary by orders of magnitude as a function of reference method, variations in equipment, and other systematics that may not be documented in all (or even any) of the included studies. By estimating both $g(x_{s,i})$ and $F_s$, simultaneously, the model was able to find the optimal balance between the shape of the function and the alignment of observations along zones of overlapping age. Finally, individual observations are assumed to vary with respect to the lifespan function given normally distributed noise $\varepsilon$ with an unknown error term $\sigma$, chosen as a minimally informative maximum entropy distribution for the error (*Jaynes, 2003*). In total, this gives the following general form for each observation in our data $y_{s,i}$:

$$y_{s,i} = g(x_{s,i}) \cdot F_s + \varepsilon, where\, \varepsilon = Normal(0.0, \sigma)$$

The scaling factor $F_s$ is intended to act as a feature-scaled standardization of the unitless function $g(x_{s,i})$. Because of this, the scaling factors were constrained in two ways. The first was a hard constraint that truncated their distributions to exclude negative values. The second was a soft constraint imposed by using a weakly regularizing prior to favor values close to unity. Since each of

the datasets had at least a few dozen subjects, the lion's share of the scaling factor's effect size was determined by the data themselves, with regularization serving mainly to keep the tails of the posterior distribution from considering unreasonably large values, effectively penalizing the likelihood of such extreme values. The net effect of the scaling factor was that the geometric mean of all values $g(x_{s,i})$ was equal to 1.0, despite the unscaled values from different studies varying dramatically in their original units. In addition to allowing different reference methods to be combined, the relative values of $F_s$ act as a study-by-study correction for any systematics that might otherwise shift one study's observations relative to any other study, including both known and unknown factors that might introduce a bias. As with all other parameters in our models, the scaling factor estimates are not point values, but rather are posterior probability distributions with a lower bound of zero but no proscribed upper bound.

This model is highly general, accommodating any function $g()$ the analyst deems appropriate. The parameters that must be estimated are one scaling factor $F_s$ for each study included in the meta-analysis, a global error term $\sigma$, and whichever parameters the function $g()$ requires to specify its shape. Because parameter estimates in any non-linear regression model necessarily covary, it is essential that all parameters be estimated simultaneously (*McElreath, 2020*). If, for example, each dataset was 'feature scaled' prior to the meta-analysis and then subsequently stitched together, any vertical shift needed to maximize the overlap of outcomes recorded at overlapping age ranges would necessarily be ad hoc and would not be able to balance the relative weight of the evidence from each study in that area of overlapping age. By this same token, it was important that all studies included in our analysis included a range of ages that overlapped with at least one other study.

We chose two models as candidates for $g()$. Our first function $g()$ was a penalized cubic basis spline model, adapting the procedure described by *Kharratzadeh, 2017*. This provided a non-linear and semiparametric estimate of GABA differences over the lifespan as described by these eight datasets. Since it does not make strong assumptions of its own, such a model requires considerable data to make good inferences. Our second candidate was a parametric function that followed the shape of a log-normal distribution, which has been chosen in the past as a way to characterize the pattern of increases in GABA in childhood, followed by a gradual decline that describes other neurophysiological changes across the lifespan (*Lebel et al., 2012*). We include this second function primarily to contrast how powerfully the choice of functions, and of their underlying assumptions, can influence the estimate.

In order to ensure that all estimates were permitted to covary appropriately, we obtained posterior probability distributions for each parameter numerically using a Bayesian paradigm (*Gelman et al., 2014*) and implemented with the Stan programming language (*Carpenter et al., 2017*). Note that a 'posterior probability distribution' is a statistical term in this context and is not intended for use in anatomical orientation. Details of these analyses can be found in the supplementary information.

## Acknowledgements

ECP was supported by NIAAA K01 AA025306 and the McKnight Brain Research Foundation; the Center for Cognitive Aging and Memory at the University of Florida. NAJP received salary support from NIMH R00 MH107719. RAEE received salary support from NIMH R01 MH106564 and R21 HD1000869. Data were provided from NIMH R01 MH106564. The published Big GABA dataset from https://www.nitrc.org/projects/biggaba/ from *Mikkelsen et al., 2017* and this analysis derives support from NIBIB R01 EB016089.

## Additional information

### Funding

| Funder | Grant reference number | Author |
| --- | --- | --- |
| National Institutes of Health | KO1AA025306 | Eric C Porges |
| Evelyn F. McKnight Brain Research Foundation | | Eric C Porges |

| UF Health | | Eric C Porges |
| National Institutes of Health | R00MH107719 | Nicolaas AJ Puts |
| National Institutes of Health | R01EB016089 | Richard AE Edden PhD |
| National Institutes of Health | R01MH106564 | Richard AE Edden PhD |

The funders had no role in study design, data collection and interpretation, or the decision to submit the work for publication.

### Author contributions

Eric C Porges, Conceptualization, Resources, Formal analysis, Funding acquisition, Methodology, Writing - original draft, Writing - review and editing; Greg Jensen, Conceptualization, Software, Formal analysis, Visualization, Methodology, Writing - original draft, Writing - review and editing; Brent Foster, Data curation, Methodology, Writing - original draft; Richard AE Edden, Resources, Methodology, Writing - review and editing; Nicolaas AJ Puts, Conceptualization, Funding acquisition, Methodology, Writing - original draft, Project administration, Writing - review and editing

### Author ORCIDs

Eric C Porges (ID) https://orcid.org/0000-0003-3885-5859
Greg Jensen (ID) https://orcid.org/0000-0001-5050-4360

### Decision letter and Author response

Decision letter https://doi.org/10.7554/eLife.62575.sa1
Author response https://doi.org/10.7554/eLife.62575.sa2

# Additional files

## Supplementary files

• Transparent reporting form

## Data availability

All data and code used in this manuscript can be found here https://osf.io/rmhwc/.

The following dataset was generated:

| Author(s) | Year | Dataset title | Dataset URL | Database and Identifier |
|---|---|---|---|---|
| Porges EC, Jensen G, Puts NA | 2020 | The trajectory of cortical GABA levels across the lifespan | https://osf.io/rmhwc/ | Open Science Framework, rmhwc |

The following previously published dataset was used:

| Author(s) | Year | Dataset title | Dataset URL | Database and Identifier |
|---|---|---|---|---|
| Mikkelsen M, Barker PB, Bhattacharyya PK, Brix MK, Buur PF, Cecil KM, Chan KL, Chen DY, Craven AR, Cuypers K, Dacko M, Duncan NW, Dydak U, Edmondson DA, Ende G, Ersland L, Gao F, Greenhouse I, Harris AD, He N, Heba S, Hoggard N, Hsu TW, Jansen JFA, Kangarlu A, Lange T, Lebel RM, | 2017 | Big GABA: Edited MR spectroscopy at 24 research sites | https://www.nitrc.org/projects/biggaba/ | NITRC, Big GABA |

Li Y, Lin CE, Liou JK, Lirng JF, Liu F, Ma R, Maes C, Moreno-Ortega M, Murray SO, Noah S, Noeske R, Noseworthy MD, Oeltzschner G, Prisciandaro JJ, Puts NAJ, Roberts TPL, Sack M, Sailasuta N, Saleh MG, Schallmo MP, Simard N, Swinnen SP, Tegenthoff M, Truong P, Wang G, Wilkinson ID, Wittsack HJ, Xu H, Yan F, Zhang C, Zipunnikov V, Zöllner HJ, Edden RAE

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
