## [Decision Letter]

**Acceptance summary:**

This meta-analysis combines GABA magnetic resonance spectroscopy (MRS) data from multiple studies to estimate the lifespan trajectory of GABA levels. The manuscript is well-motivated and clear in its aim to untangle apparent contradictions which appear in the literature of age-dependent GABA concentration trajectories. The process of combining MRS datasets from different studies acquired using different MRS protocols is extremely challenging and difficult to achieve, but the approach presented and the GABA trajectories revealed should be of broad interest.

**Decision letter after peer review:**

Thank you for submitting your article "The trajectory of cortical GABA across the lifespan: An individual participant data meta-analysis of edited MRS studies" for consideration by *eLife*. Your article has been reviewed by 3 peer reviewers, and the evaluation has been overseen by Chris Baker as the Reviewing and Senior Editor. The following individual involved in review of your submission has agreed to reveal their identity: William Clarke (Reviewer #2).

The reviewers have discussed the reviews with one another and the Reviewing/Senior Editor has drafted this decision to help you prepare a revised submission. There was vigorous discussion and while all reviewers appreciated the attempt to estimate the trajectory of GABA across the lifespan, there were many concerns raised about the difficulties of combining MRS studies.

As the editors have judged that your manuscript is of interest, but as described below that additional data and analyses are required before it is published, we would like to draw your attention to changes in our revision policy that we have made in response to COVID-19 (https://elifesciences.org/articles/57162). First, because many researchers have temporarily lost access to the labs, we will give authors as much time as they need to submit revised manuscripts. We are also offering, if you choose, to post the manuscript to bioRxiv (if it is not already there) along with this decision letter and a formal designation that the manuscript is "in revision at *eLife*". Please let us know if you would like to pursue this option. (If your work is more suitable for medRxiv, you will need to post the preprint yourself, as the mechanisms for us to do so are still in development.)

Summary:

The aim of this meta-analysis is to combine GABA MRS data from multiple studies to estimate the lifespan trajectory of GABA levels. The manuscript is well-motivated and clear in its aim to untangle apparent contradictions which appear in the literature of age-dependent GABA concentration trajectories. The methods are well written and do a superb job to explain the potentially tricky to understand IPD-MA approach. However, the process of combining MRS datasets from different studies acquired using different MRS protocols is extremely challenging and difficult to achieve.

Essential revisions:

There are three major issues the reviewers would like to see the authors address. The comments below reflect the discussion between the reviewers. In general, the reviewers would like to see much more detail and justification of the approach throughout the manuscript.

1. Exclusion of a large number of publications

The reviewers appreciate that the method used requires that for a study to be included it contains a range of ages, and that this individual subject data is available for the meta-analysis. Thus the exclusion of a large number of studies is not unexpected. However, out of 273 MRS GABA publications the authors only identify 57 as being relevant and ultimately only use data from 8.

The meta-analysis would be much stronger if more data were included. In the text it is stated that, "Of the 55 relevant studies retrieved in the systematic search, only 7 included ﬁgures suitable for data extraction or had data freely available in available online repositories." Did the authors make an attempt to contact the corresponding authors of the identified studies to try and obtain the data that was not readily available from the publications? This would also potentially allow the calculation of comparable GABA values across studies.

We would also like to see more detail regarding the reasons for exclusion of so many studies including the precise spectral quality exclusion metrics.

2. Comparability between studies

There are many differences between the included studies: voxel location (many different areas across the PFC, parietal and occipital areas), voxel size (impacting the varying portion of GM), TR (implying that the degree of T1-weighted is different across studies) and quantification including wether the water signal was fully relaxed.

The logic of the IPD-MA approach is that data collection methods only need to be consistent across all subjects in a study (i.e. internally consistent). All other differences are lumped into the single scaling factor. Even those factors that have non-linear functions (e.g. partial saturation) should only result in independent fixed scalings between datasets.

But how can we be confident that the approach is able to discount all those difference between studies? Is the method sufficient for the authors to be able to draw the conclusions they do? The authors don't give bounds on dataset scaling factors. While the method is in principal sound it may be that the single point-estimate currently shown disguises a lack of sensitivity in determining correct scaling factors.

3. Clarification on the validity of Figures 4/5

The authors should justify the suitability of the log-normal, which is not clearly explained.

How do the authors account for the clear discontinuity in Figure 4 around 20-30 years of age? It suggests that the individual scaling factors applied were ineffective and may reflect the difficulty of accounting for all the differences between studies as noted above and hence, questions the legitimacy of combining different MRS studies in this way. Is it because datasets with narrower age ranges have many reasonable/possible scaling values?

[Editors' note: further revisions were suggested prior to acceptance, as described below.]

Thank you for submitting your article "The trajectory of cortical GABA across the lifespan, an individual participant data meta-analysis of edited MRS studies" for consideration by *eLife*. Your article has been reviewed by 2 peer reviewers, and the evaluation has been overseen by Chris Baker as Reviewing and Senior Editor. The following individuals involved in review of your submission have agreed to reveal their identity: Jeffrey Stanley (Reviewer #1); William Clarke (Reviewer #2).

Essential revisions:

While the reviewers think the manuscript has improved, there are still some remaining concerns that need to be addressed as outlined in the Reviewer comments below. Please carefully consider and respond to these concerns.

*Reviewer #1:*

Overall, the revised manuscript is an improvement and more focused but still has many issues to addressed, which includes the following.

The trajectory of cortical GABA across the lifespan, an individual participant data meta-analysis of edited MRS studies

In the abstract, it is important to state how GABA is being expressed and from where in the brain with respect to the main outcome of the modelling.

In the 2nd paragraph of the introduction, it would be helpful to state whether cited studies are based on cross-sectional data or longitudinal data.

In the 3rd paragraph of the introduction, the text is confusing, verbose with repeating statements, and hence, difficult to follow.

The statement describing MEGAPRESS, "The majority of MRS studies of GABA at 3 Tesla have utilized the unique structure of the GABA molecule to selectively edit the GABA signal using a typical MRS acquisition with frequency-selective pulses", is confusing and inaccurate – the isolation of GABA is derived by taking the difference between two spectra with and without the selective saturation – technically, MEGAPRESS is not spectral editing approach but a "difference editing" approach by proper definition.

The statement, "studies have shown that the majority of the GABA measured using MRS reflects intracellular levels rather than synaptic levels", is confusing – GABA is initially defined as being localized either in the intracellular or extracellular space – where does "synaptic level" fit in? It should be viewed as part of the intracellular space.

The section, "The importance of GABA in cognition motivates an understanding of the age-relationship", in the introduction does not help to focus the narrative or help to motivate the aims of the manuscript – it should be removed.

The statement, "the GABA signal is considered to be reflective of inhibitory tone rather than dynamic synaptic inhibition", is confusing and needs clarification – "dynamic synaptic inhibition" is too convoluted.

The information in Table 1 can be extended to also include ratio or absolute, the voxel location(s) and sample size for each study.

In Figure 1, the 95% CI bands are misleading and should only cover the range of data points in each graph and not the full range of the age scale – it's inappropriate to imply that the modelling in each is generalizable across the full age range – the text also conflicts this as noted below.

In the result section, first paragraph, the conclusion from the "cursory examination" is confusing with conflicting messages. The authors are arguing that studies reporting linear differences with age are inaccurate and non-linear is better. Then note that linear trends over short intervals are OK. Also, the fact that the "linear trends explained less than 20% of the variance" does not help the author's argument whether linear or no-linear is better – the assumption that GABA must show age effects across all ages may not be correct. This needs to b better articulated.

In Figure 3, the rationale to only focus on data from the posterior is unclear. What is the significance of the posterior vs frontal data or using both? This needs to be explained.

The results to support similar age effects when expressed as ratio vs absolute is weak and not compelling. The bulk of the data is dominated by studies reporting ratios, which does not help. If the GABA age effects are not driven by age effects with PCr+Cr, then it would imply that PCr+Cr levels are not different across age, which is not the case as supported

The Figure 4 caption is confusing – it refers to models shown in Figure 4.

Regarding the main finding of a "sharp" or "rapid" early increase in GABA is driven by data from one study (weak) and results are expressed as a ratio. For example, a miss adjustment in the scaling factor on that one study could artificially enhance this effect. Also, the bulk of the data from those 8 studies expressed GABA as a ratio and GABA/PCr+Cr ratio values do not equate to GABA levels especially when PCr+Cr levels differ across age as noted in the literature (far deeper than what is noted in the text and including ratios relative to NAA does not help). Therefore, greater caution is needed in reporting this observation – a softer tone is needed including in the abstract.

In the discussion, it must be emphasized at the beginning that GABA/PCr+Cr ratio values do not equate to GABA levels or concentration – this needs to be acknowledged and corrected in the discussion – one must be careful not to over interpret the data.

In the discussion, the statement, "Cr+PCr is acquired during the same MEGAPRESS scan as GABA, limiting effects of chemical shift: Cr+PCr has a minimal shift from the GABA signal", is confusing and needs to be clarified.

The statement, "the water signal represents a more concentrated chemical yielding a higher SNR, but it may also introduce error in estimates of location due to chemical shift effects", is also confusing – unclear how chemical shift is a factor.

The discussion on the CSF correction applied in the Porges et al. 2017b completely ignores the fact that the correct was not applied correctly – it should not be treated as a co-variate in the regression but as a volume correction factor (as noted earlier in the text). It would help to address this.

*Reviewer #2:*

The authors have thoroughly addressed my one major concern, regarding the uncertainty in the per-study scaling factors at the heart of their method. The authors have introduced figure 3 to present clearly the range of potential values that each scaling factor could take. It is apparent that the overall non-linear trend identified by this work is consistent within the bounds presented in this new figure.

Furthermore, it appears the authors have engaged constructively with all reviewer comments (though I defer to my co-reviewers to determine if their comments have been addressed sufficiently). Therefore, I have no further concerns with this manuscript. I believe that it makes a substantial contribution to the field. And with the revisions, the authors are forthright in addressing the potential limitations of their work.

---

## [Author Response]

Essential revisions:There are three major issues the reviewers would like to see the authors address. The comments below reflect the discussion between the reviewers. In general, the reviewers would like to see much more detail and justification of the approach throughout the manuscript.1. Exclusion of a large number of publicationsThe reviewers appreciate that the method used requires that for a study to be included it contains a range of ages, and that this individual subject data is available for the meta-analysis. Thus the exclusion of a large number of studies is not unexpected. However, out of 273 MRS GABA publications the authors only identify 57 as being relevant and ultimately only use data from 8.The meta-analysis would be much stronger if more data were included. In the text it is stated that, "Of the 55 relevant studies retrieved in the systematic search, only 7 included ﬁgures suitable for data extraction or had data freely available in available online repositories." Did the authors make an attempt to contact the corresponding authors of the identified studies to try and obtain the data that was not readily available from the publications? This would also potentially allow the calculation of comparable GABA values across studies.

We would like to thank the reviewer(s) for drawing our attention to this. We were not sufficiently clear in the submitted version. In preparing this manuscript we used available data (either via extraction from figures where individual subject data points were presented on figures or from data repositories), and we are sharing all code used in the analysis. This approach allows for complete transparency of our analysis and potential to replicate and extend our work. If other manuscripts had otherwise met our inclusion criteria but did not provide extractable or shared data, 1) these would be incompatible with our proposed approach, and 2) the transparency and replication objectives of our previously set PRISMA framework would be violated.

We consider the more involved process of managing raw data requests from other authors to be an exciting future direction for this method and are considering pursuing it as part of a larger consortium-approach, such as http://enigma.ini.usc.edu which Dr. Porges is affiliated with. However, this is beyond the scope of the current manuscript and not in line with the PRISMA framework established at the outset.

We would also like to see more detail regarding the reasons for exclusion of so many studies including the precise spectral quality exclusion metrics.

Data were not excluded based on spectral quality, as this would not be possible to apply to extracted data given it is rarely included for all subjects. However, datasets were excluded based on acquisition methods if they did not meet field consensus for GABA quantification using MRS.

To address the above concerns of the reviewers, we have added the following text to the Data Quality Section:

“A third step was performed to assess whether studies adhered to consensus quality assurance criteria for data collection, analysis, and reporting. […] Spectral quality was not a criterion for exclusion, as individual data spectra or quality metrics (e.g., linewidth, CRLB, etc.) are rarely if ever published in manuscripts used for extraction in our IPD-MA.”

2. Comparability between studiesThere are many differences between the included studies: voxel location (many different areas across the PFC, parietal and occipital areas), voxel size (impacting the varying portion of GM), TR (implying that the degree of T1-weighted is different across studies) and quantification including wether the water signal was fully relaxed.The logic of the IPD-MA approach is that data collection methods only need to be consistent across all subjects in a study (i.e. internally consistent). All other differences are lumped into the single scaling factor. Even those factors that have non-linear functions (e.g. partial saturation) should only result in independent fixed scalings between datasets.But how can we be confident that the approach is able to discount all those difference between studies? Is the method sufficient for the authors to be able to draw the conclusions they do? The authors don't give bounds on dataset scaling factors. While the method is in principal sound it may be that the single point-estimate currently shown disguises a lack of sensitivity in determining correct scaling factors.

This is a valid concern, and we appreciate the reviewers(s) for expressing it and giving us the chance to elaborate. Using a per-study scaling factor is not a way of uncovering, or even identifying the factors that result in systematic bias in a dataset. Rather, such a factor seeks to estimate, relative to the other studies that are included, the multipliers needed to bring the included studies into alignment with one another. This has been clarified in the revision. Additionally, we now articulate explicitly that, although these scaling factors have a lower bound of zero (since negative MRS signals are a priori impossible), they have no strict upper bound. Further, we express explicitly that our scaling factor estimates, like all parameters in the model, are posterior distributions and not point-estimates. If there is a lot of uncertainty in which scaling factor is ideal, this results in a more uncertain posterior estimate of the scaling factor, as an automatic consequence of the principles of Bayesian inference.

3. Clarification on the validity of Figures 4/5The authors should justify the suitability of the log-normal, which is not clearly explained.How do the authors account for the clear discontinuity in Figure 4 around 20-30 years of age? It suggests that the individual scaling factors applied were ineffective and may reflect the difficulty of accounting for all the differences between studies as noted above and hence, questions the legitimacy of combining different MRS studies in this way. Is it because datasets with narrower age ranges have many reasonable/possible scaling values?

As we would like to incorporate this useful feedback, we now note in the manuscript that, although the *mean* of the splines shows a slight upward trend in this age range, this does not mean that every spline that is nominated by this model must include this feature. Any doubly-differentiable curve that passes through the shaded region in Figure 4 (Left) is in principle nominated by the spline model, and it would not be at all surprising if certain kinks and wrinkles in the present estimate smoothed out as additional datasets were added. What *is* expected given the model’s assumptions is that the true curve, whatever its shape, probably mostly resides within the shaded region.

[Editors' note: further revisions were suggested prior to acceptance, as described below.]

Reviewer #1:[…] In the abstract, it is important to state how GABA is being expressed and from where in the brain with respect to the main outcome of the modelling.

We now explicitly state we studied GABA from frontal regions for the meta-analysis and explicitly state that any measure of GABA (but only from editing) was accepted in the search criteria within the confines of the limited abstract length.

Explicitly, we have amended the following:

“No single study to date has included the entire lifespan. In this study, 8 suitable datasets were integrated to generate a model of the trajectory of frontal GABA levels (as reported through edited MRS; both expressed as ratios and in institutional units).”

“Integrated data show the lifespan trajectory of frontal GABA measures involves an early period of increase,[…]”

In the 2nd paragraph of the introduction, it would be helpful to state whether cited studies are based on cross-sectional data or longitudinal data.

None of the studies were longitudinal. We now explicitly state all studie were cross-sectional in the last sentence of paragraph 2:

“Of interest, all studies were cross-sectional rather than looking at within-subject change in GABA with age.”

In the 3rd paragraph of the introduction, the text is confusing, verbose with repeating statements, and hence, difficult to follow.

We thank the reviewer for this comment and have tidied this section up as below:

“However, conflicting results only exist when looking at different age-ranges and consistent findings have been shown in studies of similar age-range. […] In the absence of a lifespan study, we implemented an individual participant data meta-analytic (IPD-MA) approach following PRISMA guidelines (Moher, Liberati, Tetzlaff, and Altman, 2009; see Figure 5) supplemented with data collected by the authors and previously published in summary form (Puts et al., 2017).”

The statement describing MEGAPRESS, "The majority of MRS studies of GABA at 3 Tesla have utilized the unique structure of the GABA molecule to selectively edit the GABA signal using a typical MRS acquisition with frequency-selective pulses", is confusing and inaccurate – the isolation of GABA is derived by taking the difference between two spectra with and without the selective saturation – technically, MEGAPRESS is not spectral editing approach but a "difference editing" approach by proper definition.

We have rephrased this section and have expressed a clearer introduction to MEGA-PRESS as a J-difference editing technique. We have also re-ordered the paragraph to enhance flow. However, we based our description of that of the MRS of GABA consensus (Choi et al. NMR Biomed. 2020) which states the definition as follows:

“Spectral editing in in vivo 1H‐MRS provides an effective means to measure low‐concentration metabolite signals that cannot be reliably measured by conventional MRS techniques due to signal overlap, for example, γ‐aminobutyric acid, glutathione and D‐2‐hydroxyglutarate. Spectral editing strategies utilize known *J*‐coupling relationships within the metabolite of interest to discriminate their resonances from overlying signals." under which definition this would be considered spectral editing.

Our revised introduction now states the following:

“The majority of MRS studies of GABA at 3 Tesla have utilized J-difference editing to selectively ‘edit’ the GABA signal (e.g. Mullins et al., 2014). […] This limitation constrains the spatial specificity of the measurement to coarse regions that often lack discrete functional specificity.”

The statement, "studies have shown that the majority of the GABA measured using MRS reflects intracellular levels rather than synaptic levels", is confusing – GABA is initially defined as being localized either in the intracellular or extracellular space – where does "synaptic level" fit in? It should be viewed as part of the intracellular space.

We thank the reviewer for this comment. We have rewritten this section for clarity which now reads:

“The functional relevance of MRS measures of GABA is important to note. Measured GABA levels include both intracellular (both somatic and synaptic) and extracellular contributions to the overall GABA concentration. […] GABA levels measured via MRS in young adults at rest have been reported to be stable for up to 7 months (Near et al., 2014) and do not exhibit a diurnal rhythm (Evans et al., 2010).”

The section, "The importance of GABA in cognition motivates an understanding of the age-relationship", in the introduction does not help to focus the narrative or help to motivate the aims of the manuscript – it should be removed.

In the nature of transparency, this in fact did motivate our study initially, given we have a strong interest in lifespan-related and GABA-dependent changes in cognition and sensory processing. We are therefore reluctant to remove this statement (since it does in fact motivate this study) but we agree it was overly lengthy. We have substantially reduced this paragraph and the next, to emphasize that age-related changes are important to account for in studies of the functional role of GABA. It now reads:

**“**The importance of GABA in the context of ageing in health and disease

GABA levels measured with MRS have been linked to clinical and cognitive outcomes. […] Our approach allows for a systematic review of existing work studying GABA across development and aging with immediate impact on future studies in the context of development and aging.”

The statement, "the GABA signal is considered to be reflective of inhibitory tone rather than dynamic synaptic inhibition", is confusing and needs clarification – "dynamic synaptic inhibition" is too convoluted.

As per our fifth response, this section was revised including this statement. We now refer to the interpretation of Rae (2014).

The information in Table 1 can be extended to also include ratio or absolute, the voxel location(s) and sample size for each study.

Table 1 refers to the regression coefficients (in the results first, as per *eLife* guidance). However, we have added the text:

“Further study details including sample sizes can be found in Tables 2 and 3.”

In Table 2, which references acquisition and analysis details, ratio was previously reported and is now detailed further for clarity. Voxel location has been added to Table 2.

Sample sizes were already included in the Table 3 and we believe they fit better there. The following text has been added to the Table 2 captions:

“Reference method refers to either reference to water (in estimated concentration / H_2_O) or as a ratio to PCr + Cr (Cr+PCr) and describes whether data was acquired macromolecule-suppressed (GABA) or as GABA GABA+ macromolecules (GABA+).”

In Figure 1, the 95% CI bands are misleading and should only cover the range of data points in each graph and not the full range of the age scale – it's inappropriate to imply that the modelling in each is generalizable across the full age range – the text also conflicts this as noted below.

We apologize for the confusion and fully agree with the Reviewer. Figure 1 explicitly shows what *not* to do, which is extrapolate beyond the data, as a visualization of what many studies *do* end up doing. Here we show that these data are not generalizable. In response, we 1) clarify the goal of this figure and address that generalizability is inappropriate. 2) We also rephrase the last sub figure as “extrapolated aggregate” and 3) We have the figure to reflect the correlation that covers the sample and which that does not by making the 95%CI transparent beyond the sample of the data.

Figure caption 1 now reads:

“Figure 1: Linear relationships between age and GABA signal, showing that linear extrapolation over the lifespan is not appropriate. In each dataset, GABA was scaled relative to the geometric mean. Linear models were fit for each dataset separately. Dark shaded regions represent the 95% credible interval for the interpolated regression line, given the data from each study and the assumption of a linear effect, whereas the light shaded regions represent the 95% credible interval for the extrapolated regression line.”

The text in the first paragraph of the Results section now reads:

“Furthermore, assuming a linear trend is unsuccessful even at describing the data within each study. […] [PN1] Although a linear trend may provide an an approximate summary over short periods of time, a linear trend over the entire lifespan is not appropriate.”

In the result section, first paragraph, the conclusion from the "cursory examination" is confusing with conflicting messages. The authors are arguing that studies reporting linear differences with age are inaccurate and non-linear is better. Then note that linear trends over short intervals are OK. Also, the fact that the "linear trends explained less than 20% of the variance" does not help the author's argument whether linear or no-linear is better – the assumption that GABA must show age effects across all ages may not be correct. This needs to be better articulated.

We thank the reviewer for this comment and have clarified this section, also with respect to the figure discussed in our ninth response. It is now phrased as:

“Furthermore, assuming a linear trend is unsuccessful even at describing the data within each study. […] [PN1] Although a linear trend may provide an approximate summary over short periods of time, a linear trend over the entire lifespan is not appropriate.”

In Figure 3, the rationale to only focus on data from the posterior is unclear. What is the significance of the posterior vs frontal data or using both? This needs to be explained.

Figure 3 refers to the Bayesian posterior probability distributions for parameter estimates; that is, parameter estimates that have taken both the prior and the likelihood into account (since posterior = (likelihood * prior)/constant). We have clarified our use of this statistical term, which coincidentally shares a name with the anatomical region, in both the figure legend and in the body of the manuscript.

With respect to Figure 4 and Supplementary Figure 1 (which does refer to Frontal and Posterior cortex measures); this was in response to a prior reviewer comment who wanted us to also show data for frontal-only data only given that we focus on frontal regions but also include posterior regions where possible. This reviewer was concerned about the potential impact of posterior regions due to regional differences. We now justify this figure more clearly in the text.

“We additionally also explored the potential effect of region by performing the analysis on only frontal data (removing Mikkelsen et al. and Simmonite et al. respectively) with no substantial impact on the direction of the slope over time. This information can be found in Supplemental figure 1.”

The results to support similar age effects when expressed as ratio vs absolute is weak and not compelling. The bulk of the data is dominated by studies reporting ratios, which does not help. If the GABA age effects are not driven by age effects with PCr+Cr, then it would imply that PCr+Cr levels are not different across age, which is not the case as supported

We agree that variability in the methods aggregated in this IPD-MA are a limitation but we can only rely on the available data with respect to comparison between ratio versus “absolute” measures of GABA. Further; we are aware of, and have enriched the discussion of, the work attempting to characterize age related changes in PCr+Cr during aging. The most common, but certainly not universal results, have shown an age related increase in PCr+Cr with a great deal of variability.

While we can’t actually comment on whether */Cr+PCr* or water referenced data are consistent with each other, we do have overlapping data available for Gao et al. and Porges et al. and these do support similar age effects across methods. Further we report that when looking at overlapping age ranges from the two studies, there is no significant difference in an Age x Gaba correlation between the two. Based on the reviewer comment we have limited and softened our interpretation of a non-significant result (P17), but it does not support the notion that age-related increases in PCr+Cr are driving the age-related decrease in GABA which is discussed in the following text (P19).

“Furthermore, a limited effect of tissue is also suggested by consistency between findings of GABA+/Cr+PCr and estimated concentrations. There is no significant difference in the correlation for GABA x age in the overlapping age range of 43 to 76 years between Porges et al. (GABA+/H2O) and Gao et al. (GABA+/Cr+PCr) who looked at comparable samples (Fisher R-to-Z; (z = 1.48, p = 0.19)) showing consistency between studies focusing on the same population (see also the discussion above).”

A 2019 systematic review (Cleeland et al. 2019) provides mixed evidence that generally supports an age related increase in */Cr+PCr*. Since the publication of that review, a 2020 study at 7T has produced further data suggesting a modest increase of */Cr+PCr* in some but not all brain regions during aging (Lind et al. J. Neurosci 2020).

Importantly, an age related increase in */Cr+PCr* would only *enhance* the GABA-effect seen in aging as we report in the discussion. No comparable data are available in early development.

Thus there is indeed some, but limited, support that this is driven by GABA and not the reference compound and we are now more careful in interpreting this as strong evidence and discuss this further in the discussion (see also our fourteenth response). We have now modified the section on page 18 to read:

“The impact of increasing Cr+PCr during aging would be to make the any age-related decrease in GABA/Cr+PCr more pronounced, and the potential of this to contribution should not be ruled out. […] Our leave-one-out approach to the ‘late-age’ data (Figure 4) found no substantial differences in the final models, providing some support to the notion that differences in Cr+PCr are not driving the GABA findings presented here but the impact of reference compound needs to be considered.”

As is noted in our fourteenth response, we also recommend reporting of both ratios and estimated concentrations when possible.

The Figure 4 caption is confusing – it refers to models shown in Figure 4.

We now ensure figure captions match the right figures. We have renamed the Supplemental Figure to Supplemental Figure 1 and explain this accompanies Figure 4 in the main manuscript.

Regarding the main finding of a "sharp" or "rapid" early increase in GABA is driven by data from one study (weak) and results are expressed as a ratio. For example, a miss adjustment in the scaling factor on that one study could artificially enhance this effect. Also, the bulk of the data from those 8 studies expressed GABA as a ratio and GABA/PCr+Cr ratio values do not equate to GABA levels especially when PCr+Cr levels differ across age as noted in the literature (far deeper than what is noted in the text and including ratios relative to NAA does not help). Therefore, greater caution is needed in reporting this observation – a softer tone is needed including in the abstract.

We have softened the language on a sharp increase in development in both the main body of the manuscript and in the text and removed any reference to ‘rapid’. With respect to the ratios, our approach is designed to handle differences in ratio and we have added a substantial paragraph (see below). It should be noted that at least in one of our recent papers, correlations between GABA/Glx and correlations hold between GABA IU and GABA*/Cr+PCr*. We now are more explicit (see also our twelfth response) in detailing this as a limitation of our meta-analysis and of the literature and recommend either reporting both GABA IU and GABA*/Cr+PCr* unless authors have clear expectations/justification regarding reporting of their data.

Our clarifying language in the discussion now reads:

“One of the cornerstone assumptions of our analysis is that, although the use of different references results in different fundamental units, participants within any given study using a given method can be compared to one another in relative terms. […] Provided this assumption is met, however, simultaneous estimation of between-list scaling factors and within-list participant differences provides a robust means of integrating multiple studies into a single overall trajectory.”

And we also added the following:

“Future studies should provide concentration values across the lifespan; we recommend reporting of values both as metabolite as well as water ratios to look at within-study consistency and robustness of the effect of GABA. Recent developments in MRS of GABA methodology will allow for this, even when data is collected on multiple scanner platforms (Saleh et al., 2019).”

In the discussion, it must be emphasized at the beginning that GABA/PCr+Cr ratio values do not equate to GABA levels or concentration – this needs to be acknowledged and corrected in the discussion – one must be careful not to over interpret the data.

We agree with the reviewer and incorrectly generalized these findings to “GABA levels” in the discussion and are more careful throughout the manuscript. We now refer to these as GABA “measures” and explicitly state that we included studies in UI and as ratios and as ‘estimates’ elsewhere. See also our fourteenth response.

In the discussion, the statement, "Cr+PCr is acquired during the same MEGAPRESS scan as GABA, limiting effects of chemical shift: Cr+PCr has a minimal shift from the GABA signal", is confusing and needs to be clarified.

We have clarified this statement to state:

“For instance, the Cr+PCr at 3.05 ppm has minimal chemical shift from the GABA signal at 3 ppm (Mullins et al. 2014) and is acquired during the MEGA-PRESS sequence. […] In contrast, the water signal represents a more concentrated chemical yielding a higher SNR, but it may also introduce error in estimates of location due to chemical shift effects (Mullins et al., 2014) when not acquired from the same voxel as GABA and on some scanners requires a separate acquisition (see Choi et al. for a discussion).”

We now also refer to prior discussions on reference compound that are beyond our paper.

The statement, "the water signal represents a more concentrated chemical yielding a higher SNR, but it may also introduce error in estimates of location due to chemical shift effects", is also confusing – unclear how chemical shift is a factor.

We now clarify this statement (see our sixteenth response). As discussed in the various consensus papers that came out over the past 2 years, acquisition of a water unsuppressed scan brings in several difficulties. E.g. sometimes water is acquired from the same voxel (where the voxel is centred at 3 ppm) thus water is acquired from outside of the voxel that GABA was collected in. As per our sixteenth response, this is beyond the scope of this manuscript and we now refer to Mullins et al. 2014 and Choi et al. 2020.

The discussion on the CSF correction applied in the Porges et al. 2017b completely ignores the fact that the correct was not applied correctly – it should not be treated as a co-variate in the regression but as a volume correction factor (as noted earlier in the text). It would help to address this.

In Porges et al. 2017b, which is a study of the impact of different tissue-correction approaches, CSF correction was applied as volume correction, not as part of the regression model. Additionally α-corrected tissue correction proposed by Harris et al. was applied. We now clarify the impact of CSF volume correction in the manuscript:

“In water-referenced studies of age-related differences in older cohorts (where bulk tissue changes are most likely to impact this relationship), accounting for cerebrospinal fluid (CSF) in the tissue-correction approach does not remove a significant relationship between age and cortical GABA (Porges et al., 2017b).”

In Porges et al. 2017a CSF was part of the regression model, but this does not impact the analysis performed here, as individual data points were extracted from that study, no data from the regression model were used.